# Understanding Programmatic Weak Supervision via Source-aware Influence Function

**Jieyu Zhang**[1]*, **Haonan Wang**[2]*, **Cheng-Yu Hsieh**[1], **Alexander Ratner**[1,3]
[1]University of Washington    [2]University of Illinois Urbana-Champaign    [3]Snorkel AI, Inc.
{jieyuz2, cydhsieh, ajratner}@cs.washington.edu
{haonan3}@illinois.edu

## Abstract

Programmatic Weak Supervision (PWS) aggregates the source votes of multiple weak supervision sources into probabilistic training labels, which are in turn used to train an end model. With its increasing popularity, it is critical to have some tool for users to understand the influence of each component (*e.g.*, the source vote or training data) in the pipeline and interpret the end model behavior. To achieve this, we build on Influence Function (IF) and propose source-aware IF[2], which leverages the generation process of the probabilistic labels to decompose the end model's training objective and then calculate the influence associated with each (data, source, class) tuple. These primitive influence score can then be used to estimate the influence of individual component of PWS, such as source vote, supervision source, and training data. On datasets of diverse domains, we demonstrate multiple use cases: (1) interpreting incorrect predictions from multiple angles that reveals insights for debugging the PWS pipeline, (2) identifying mislabeling of sources with a gain of 9%-37% over baselines, and (3) improving the end model's generalization performance by removing harmful components in the training objective (13%-24% better than ordinary IF).

## 1 Introduction

One of the major bottlenecks for deploying modern machine learning models is the need for a substantial amounts of human-labeled training data, often annotated over long periods of time and at great expense. As machine learning models become increasingly powerful but also data hungry, new "data-centric" AI development workflows and systems have emerged, wherein the labeling and development of this training data is positioned and supported as the central development activity. One recent and increasingly popular type of data-centric AI development uses **Programmatic Weak Supervision (PWS)**, wherein users focus on developing a diversity of noisy, programmatic supervision sources [32, 33, 50, 46] to programmatically annotate training data in an efficient way. Specifically, these *weak supervision sources*, *e.g.*, heuristics, knowledge bases, and pre-trained models, are often abstracted as *labeling functions (LFs)* [33], which is a user-defined program that provides potentially noisy labels for some subset of the data. So far, different modeling techniques have been developed to aggregate the noisy votes of LFs to produce training labels (often referred to as a *label model*) [33, 31, 14, 40]. Finally, these training labels are in turn used to train an *end model* for the downstream classification tasks. In this study, we focus on this *two-stage* PWS pipeline [46].

With the increasing popularity of PWS in various applications [13, 36, 40, 19], it is of great importance to provide practitioners with a tool that helps them understand the behavior of a trained end model

---

*These authors contributed equally to this work.

[2]The implementation is avaiable at the WRENCH benchmark:https://github.com/JieyuZ2/wrench

36th Conference on Neural Information Processing Systems (NeurIPS 2022).

as an effect of each upstream component (*e.g.*, source vote, training data, LFs, *etc.*) involved in the model's training process. First, by understanding what are the deciding factors that lead to a model's specific prediction, users are able to verify whether such influence is desirable in terms of critical aspects such as model safety and fairness [11, 35, 16]. Secondly, since developing LFs is still inevitably a partly manual process in real-world applications and may involve human expert in an iterative process [5, 20, 51], it is beneficial to offer users with feedback of how individual component (like each LF) influences the end model performance on the downstream task. Finally, the rendered understanding of (in)efficacy of the PWS pipeline can be naturally exploited to automatically improve the end model performance.

Towards the goal of developing tools to help understand the model behavior, Influence Function (IF) [22, 15, 8] has been recently applied to interpret the model prediction as influence of the training data for large-scale applications. However, most of existing IF studies are unaware of the generation process of training labels like PWS and only estimate the influence of the training data [22, 3]. Notably, the group IF study [23] has attempted to apply IF on PWS, but it simply estimates the influence of each LF as summation of data influence and is limited to a specified PWS without learnable label model. In this work, we aim to develop a general framework that helps estimate the influence of each PWS's component on the end model and is aware of (yet computationally agnostic to) the choice of label model. To achieve this, we leverage the knowledge of how the probabilistic labels are synthesized from sources and accordingly decompose the training loss of each data into multiple terms, each of which is associated with what we called $(i, j, c)$-*effect*, *i.e.*, the effect of $j$-th LF on $c$-th dimension (corresponding to class $c$) of the $i$-th data's probabilistic label. With such a training loss decomposition, we propose two means of calculating our *source-aware IF*, namely, reweighting and weight-moving, based on different perturbations on the training respectively. Each estimated source-aware influence score can be interpreted as the influence of removing the $(i, j, c)$-effect from the pipeline. By simply aggregating over specific dimension(s), it can be utilized to estimate the influence of different components of PWS including (1) each source vote, (2) each LF and (3) each training data, since the influence score is additive [23].

In experiments, we exploit the source-aware IF in a diversity of 13 classification datasets including tabular, text, and image data, and present multiple applications. First, we show that source-aware IF can help to understand the behavior of the end model in terms of the PWS components; in particular, when interpreting the same incorrect prediction made by two PWS pipelines with different label models, data-level IF might lead to the same influential training data, whereas source-aware IF is capable of revealing the most responsible LF or source vote. Second, we use source-aware IF as a tool to identify mislabeling of each LF and show that it outperforms baselines with a significant margin of 9%-37%. Finally, we demonstrate that the training loss decomposition and source-aware IF enable fine-grained training loss perturbation and consequently lead to better test loss improvement (13%-24% improvement over ordinary IF [22] and group IF [23]).

To summarize, this work makes the following contributions. **First**, we propose source-aware IF tailored for PWS that incorporates our knowledge of the generation process of training labels. **Second**, we develop two means of calculating the source-aware IF based on different training loss perturbations. **Third**, we demonstrate a variety of use cases of the source-aware IF and conduct extensive quantitative and qualitative experiments on real datasets in different domains to verify the efficacy of source-aware IF on understanding and improving PWS pipelines.

## 2 Preliminary

We denote scalars and generic items as lowercase letters, vectors as lowercase bold letters, and matrices as bold uppercase letters. We use superscripts to index generic items, *e.g.*, the $i$-th data $x^{(i)}$, and use subscripts to index certain dimensions of a vector, *e.g.*, the $i$-th component of a vector $\mathbf{v}_i$, or the vector output of a function $\mathbf{f}(\cdot)_i$. All derivation details are deferred to the appendix.

**Programmatic Weak Supervision.** We target at a $C$-way classification problem and have a training dataset of size $N$, *i.e.*, $\mathcal{D} = \{x^{(i)}, y^{(i)}\}_{i \in [N]}$, where $x^{(i)} \in \mathcal{X} = \mathbb{R}^d$ and $y^{(i)} \in \mathcal{Y} = [C]$. The ground truth $y^{(i)}$s are not observed. We have $M$ *labeling functions (LFs)* $\{\lambda^{(j)}(\cdot)\}_{j \in [M]}$. Each $\lambda^{(j)}(\cdot) \in \{-1\} \cup \mathcal{Y}^{(j)}$, where $\mathcal{Y}^{(j)} \subset \mathcal{Y}$. That is, each $\lambda^{(j)}(\cdot)$ either abstains $(-1)$ or outputs a specific label $y \in \mathcal{Y}^{(j)}$ on a given data point. We define the label matrix as $\mathbf{L}_{ij} = \lambda^{(j)}(x^{(i)})$,

and with a slight abuse of notation, use $\mathbf{L}(x^{(i)})$ to represent the row of $\mathbf{L}$ corresponding to all of the labels assigned to $x^{(i)}$. The goal of Programmatic Weak Supervision (PWS) is to train an *end classifier* $h_\theta : \mathcal{X} \to \mathcal{Y}$ with $\{x^{(i)}\}_{i \in [N]}$ and $\mathbf{L}$ only, where $h_\theta(x) = \arg\max_{c \in [C]} \mathbf{f}_\theta(x)_c$ and $\mathbf{f}_\theta(x) \in \mathbb{R}^C$ is a vector-valued function parameterized by $\theta$. A typical PWS pipeline involves a *label model* $\mathbf{g_W}(\mathbf{L}(x))$ parametrized by $\mathbf{W}$, which aims to produce a *weak probabilistic label* $\hat{\mathbf{y}}(x^{(i)}) = \mathbf{g_W}(\mathbf{L}(x^{(i)})) \in \Delta^C$ for each data point $x^{(i)}$ (denoted as $\hat{\mathbf{y}}^{(i)}$ for simplicity). The label model produces a dataset $\hat{\mathcal{D}} = \{x^{(i)}, \hat{\mathbf{y}}^{(i)}\}_{i \in [N]}$, which is then used to train $\mathbf{f}_\theta$ via the *noise-aware loss* [33] $\hat{\ell}(\hat{\mathbf{y}}^{(i)}, \mathbf{f}_\theta(x^{(i)})) = \mathbb{E}_{y \sim \hat{\mathbf{y}}^{(i)}}[\ell(y, \mathbf{f}_\theta(x^{(i)})] = \sum_{c=1}^{C} \hat{\mathbf{y}}_c^{(i)} \ell(c, \mathbf{f}_\theta(x^{(i)}))$, where $\ell(\cdot, \cdot)$ is the cross-entropy loss, leading to the following training objective:

$$\theta^\star = \arg\min_\theta \hat{\mathcal{L}}(\hat{\mathcal{D}}; \theta) = \arg\min_\theta \frac{1}{N} \sum_{i=1}^{N} \hat{\ell}(\hat{\mathbf{y}}^{(i)}, \mathbf{f}_\theta(x^{(i)})) = \arg\min_\theta \frac{1}{N} \sum_{i=1}^{N} \sum_{c=1}^{C} -\hat{\mathbf{y}}_c^{(i)} \log(\mathbf{f}_\theta(x^{(i)})_c), \quad (1)$$

where $\theta^\star$ is the minimizer of the loss $\hat{\mathcal{L}}(\hat{\mathcal{D}}; \theta)$ over the training set $\hat{\mathcal{D}}$.

**Influence Function.** The Influence Function (IF) estimate how the minimizer $\theta^\star$ of the empirical loss $\ell$ would change if we were to reweight the $i$-th training example $z^{(i)} = (x^{(i)}, y^{(i)})$ by $\epsilon_i$. The key idea is to make a first-order Taylor approximation of the changes in $\theta^\star$ around $\epsilon_i = 0$. Specifically, if the $i$-th training sample is upweighted by a small $\epsilon_i$, the perturbed risk minimizer $\theta_{\epsilon_i}^\star$ becomes:

$$\theta_{\epsilon_i}^\star \triangleq \arg\min_{\theta \in \Theta} \frac{1}{N} \sum_{i'=1}^{N} \ell\left(y^{(i')}, \mathbf{f}_\theta(x^{(i')})\right) + \epsilon_i \, \ell\left(y^{(i)}, \mathbf{f}_\theta(x^{(i)})\right). \quad (2)$$

The changes of the model parameters due to the introduction of the weight $\epsilon_i$ are:

$$\theta_{\epsilon_i}^\star - \theta^\star \approx \left.\frac{d\theta_{\epsilon_i}^\star}{d\epsilon_i}\right|_{\epsilon_i=0} \epsilon_i = -\mathbf{H}_{\theta^\star}^{-1} \nabla_\theta \ell\left(y^{(i)}, \mathbf{f}_{\theta^\star}(x^{(i)})\right) \epsilon_i, \quad (3)$$

where $\mathbf{H}_{\theta^\star} = \frac{1}{N} \sum_{i'} \nabla_\theta^2 \ell\left(y^{(i')}, \mathbf{f}_{\theta^\star}(x^{(i')})\right)$ is the Hessian of the objective at $\theta^\star$ and assumed to be positive definite [22]. Similarly, *the change of loss with respect to a single sample* $z' = (x', y')$ is

$$\phi_i(z') = \frac{d\left(\ell(y', \mathbf{f}_{\theta_{\epsilon_i}^\star}(x')) - \ell(y', \mathbf{f}_{\theta^\star}(x'))\right)}{d\epsilon_i} = \left.\frac{d\ell(y', \mathbf{f}_{\theta_{\epsilon_i}^\star}(x'))}{d\epsilon_i}\right|_{\epsilon_i=0} \quad (4)$$
$$= -\nabla_\theta \ell\left(y', \mathbf{f}_{\theta^\star}(x')\right)^\top \mathbf{H}_{\theta^\star}^{-1} \nabla_\theta \ell(y^{(i)}, \mathbf{f}_{\theta^\star}(x^{(i)})).$$

Then, as shown in previous work [24], *the change of loss over a holdout set*, *e.g.*, the validation set $\mathcal{D}_v$, is simply the sum of the changes over every sample in the set:

$$\phi_i(\mathcal{D}_v) = \mathcal{L}(\mathcal{D}_v; \theta_{\epsilon_i}^\star) - \mathcal{L}(\mathcal{D}_v; \theta^\star) = \frac{1}{|\mathcal{D}_v|} \sum_{z' \in \mathcal{D}_v} \phi_i(z'). \quad (5)$$

The change of loss w.r.t. one sample is often interpreted as its effect on model predictions over the training set [22], while the change of loss over a holdout set can be leveraged to improve the test loss by discarding/downweighting the training examples with negative impact [38, 42, 24]. Considering the extraordinary influence that might be caused by mislabeled examples, the relative influence function (RelatIF) [3] has been proposed to measure the influence of a sample relative to its global effects. The RelatIF can be defined as $\phi_i'(z') = \frac{\phi_i(z')}{\sqrt{\phi_i(z^{(i)})}}$.

## 3  Methodology

Given an unlabeled training set $\{x^{(i)}\}_{i \in [N]}$ and $M$ labeling functions (LFs) $\{\lambda^{(j)}(\cdot)\}_{j \in [M]}$, we study the scenario where a Programmatic Weak Supervision (PWS) pipeline is employed to synthesize probabilistic labels $\{\hat{\mathbf{y}}^{(i)}\}_{i \in [N]}$ for training an end model $\mathbf{f}_\theta(x)$ for a $C$-way classification problem. We aim for a general framework for explaining and debugging the PWS pipeline through Influence Function (IF) [22], which estimates the influence of each training data on test loss/prediction for a better understanding of model behavior and is able to improve/debug the training data accordingly. In the PWS pipeline, we have full knowledge of how training labels are synthesized from different sources, *i.e.*, LFs, which opens the opportunity to trace the influence in a source-aware way. In this section, we propose the source-aware IF tailored for PWS and describe its applications.

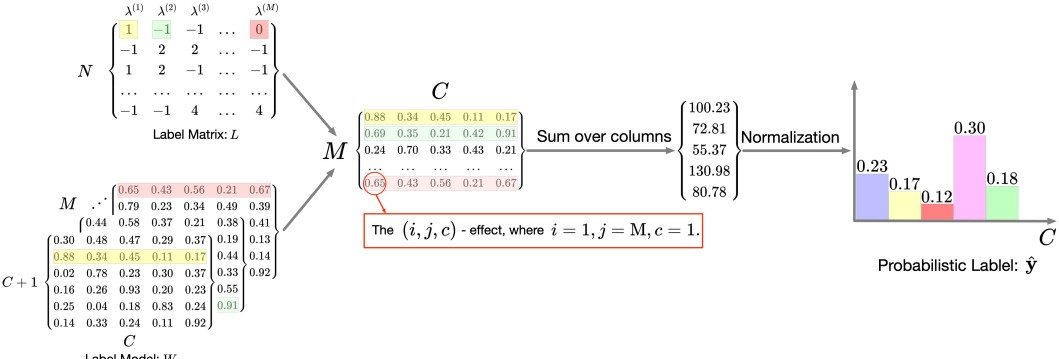

Figure 1: Illustrations of the generation process of the probabilistic training label and the $(i, j, c)$-effect. We demonstrate how the probabilistic training label for the first data point is generated from the label matrix (source votes, in other words) and the label model.

### 3.1 Training loss decomposition

By examining the popular choices of label models $\mathbf{g_W}(\cdot)$ [9, 33, 31], we show that their parameters can be unifiedly represented as a tensor $\mathbf{W} \in \mathbb{R}^{M \times (C+1) \times C}$, where the middle $(C+1)$ dimension size is due to the additional "abstention" label LFs output. The generation process of the probabilistic labels can then be expressed as

$$\forall c \in [C], \qquad \hat{\mathbf{y}}_c^{(i)} = \frac{\sigma(\sum_{j=1}^{M} \mathbf{W}_{j, \mathbf{L}_{ij}, c})}{\sum_{k=1}^{C} \sigma(\sum_{j=1}^{M} \mathbf{W}_{j, \mathbf{L}_{ij}, k})}, \tag{6}$$

where $\sigma(\cdot)$ could be (1) the identity function $\sigma_{id}(x) = x$ for the classic Majority Voting label model or its variants, or (2) the exponential function $\sigma_{exp}(x) = \exp(x)$ to recover the softmax function used in statistical models such as Dawid-Skene model [9] and Snorkel MeTaL [31]. Intuitively, label models aim to learn a source-specific weight for each LF to aggregate the votes in an unsupervised manner. Details can be found in Appendix A. In both cases, for the $i$-th data, a single parameter $\mathbf{W}_{j, \mathbf{L}_{ij}, c}$ controls the effect of the $j$-th LF's vote $\mathbf{L}_{ij}$ on the $c$-th dimension of the probabilistic label, *i.e.*, $\hat{\mathbf{y}}_c^{(i)}$. For convenience, we denote the effect of $\mathbf{L}_{ij}$ on $\hat{\mathbf{y}}_c^{(i)}$ as the $(i, j, c)$-*effect*, as it uniquely corresponds to the $i$-th data, the $j$-th LF and $c$-th dimension of the aggregated probabilistic label. For a better understanding, we visualize how a probabilistic training label is generated and the $(i, j, c)$-*effect* in Figure 1.

With such unified formulation of probabilistic label generation, we rewrite the *noise-aware loss* [33] as

$$\hat{\ell}(\hat{\mathbf{y}}^{(i)}, \mathbf{f}_\theta(x^{(i)})) = -\sum_{c=1}^{C} \frac{\sigma(\sum_{j=1}^{M} \mathbf{W}_{j, \mathbf{L}_{ij}, c})}{\sum_{k=1}^{C} \sigma(\sum_{j=1}^{M} \mathbf{W}_{j, \mathbf{L}_{ij}, k})} \log(\mathbf{f}_\theta(x^{(i)})_c). \tag{7}$$

We proceed with the $\sigma(\cdot)$ being the identity function because it enables the loss decomposition introduced below. We will later discuss other choices of $\sigma(\cdot)$. By simple rearrangement, we could decompose the loss over the training set into the *summation* of all $N \times M \times C$ terms:

$$\hat{\mathcal{L}}(\hat{\mathcal{D}}; \theta) = \frac{1}{N} \sum_{i=1}^{N} \sum_{j=1}^{M} \sum_{c=1}^{C} \bar{\ell}_{i,j,c}(\theta), \quad \bar{\ell}_{i,j,c}(\theta) = -\frac{\mathbf{W}_{j, \mathbf{L}_{ij}, c}}{\sum_{k=1}^{C} \sum_{j=1}^{M} \mathbf{W}_{j, \mathbf{L}_{ij}, k}} \log(\mathbf{f}_\theta(x^{(i)})_c). \tag{8}$$

Such a decomposition is *source-aware* in the sense that it preserves how the source votes are aggregated via the label model, and each $\bar{\ell}_{i,j,c}(\theta)$ indeed corresponds to the $(i, j, c)$-effect.

### 3.2 Source-aware Influence Function

As a consequence of training loss decomposition, we are now able to compute the influence score for each term $\bar{\ell}_{i,j,c}(\theta)$, which is the influence of removing $\bar{\ell}_{i,j,c}(\theta)$ or, in other words, the $(i, j, c)$-effect from training. We provide two means of calculating the influence score: **reweighting** and **weight-moving**, which correspond to two different ways of perturbing the training loss. Specifically, given

an (unseen) instance $z' = (x', y')$ and the cross-entropy loss $\ell(y', \mathbf{f}_\theta(x'))$, we are interested in the influence score defined by the change of loss $\ell(y', \mathbf{f}_\theta(x'))$ caused by the differences in the model parameters $\theta$ after perturbing the training loss $\hat{\mathcal{L}}(\hat{\mathcal{D}}; \theta)$. We first give the formulation of an ordinary IF for a training data $z^{(i)} = (x^{(i)}, \hat{\mathbf{y}}^{(i)})$ in the context of PWS, which is smilar to Eq. 4:

$$\phi_i(z') = -\nabla_\theta \ell\left(y', \mathbf{f}_{\theta^\star}(x')\right)^\top \mathbf{H}_{\theta^\star}^{-1} \nabla_\theta \hat{\ell}(\hat{\mathbf{y}}^{(i)}, \mathbf{f}_{\theta^\star}(x^{(i)})). \tag{9}$$

Here, $\mathbf{H}_{\theta^\star}^{-1} = \frac{1}{N} \sum_{i=1}^N \nabla_\theta^2 \hat{\ell}(\hat{\mathbf{y}}^{(i)}, \mathbf{f}_{\theta^\star}(x^{(i)}))$ and $\theta^\star$ is the minimizer of the training loss before the perturbation. We reuse the notation of $\mathbf{H}_{\theta^\star}$ and $\theta^\star$ in this section as their meanings are unchanged.

**Reweighting.** To study the influence of removing the $(i, j, c)$-effect, we reweight $\bar{\ell}_{i,j,c}(\theta)$ such that

$$\hat{\mathcal{L}}_{\epsilon_{i,j,c}}^{rw}(\hat{\mathcal{D}}; \theta) = \frac{1}{N} \sum_{i'=1}^N \sum_{j'=1}^M \sum_{c'=1}^C \bar{\ell}_{i',j',c'} + \epsilon_{i,j,c} \bar{\ell}_{i,j,c}(\theta). \tag{10}$$

Then, the influence score, *i.e.*, the difference of loss on $z' = (x', y')$ caused by reweighting the loss term $\bar{\ell}_{i,j,c}(\theta)$, is,

$$\bar{\phi}_{i,j,c}^{rw}(z') = -\nabla_\theta \ell\left(y', \mathbf{f}_{\theta^\star}(x')\right)^\top \mathbf{H}_{\theta^\star}^{-1} \nabla_\theta \bar{\ell}_{i,j,c}(\theta^\star). \tag{11}$$

In contrast to ordinary IF that reweights the loss w.r.t. individual training data, the proposed one is finer-grained since it reweights the decomposed loss terms. By reweighting, the resultant probabilistic training label is no longer sum-to-one and the solution (Eq. 11) is only for identity $\sigma(\cdot)$ function.

**Weight-moving.** We also explore an alternative way of perturbing the training loss called weight-moving. Different from reweighting, its computation is agnostic to the $\sigma(\cdot)$ function and the probabilistic training label is still sum-to-one. Specifically, we remove the $(i, j, c)$-effect in the probabilistic training label $\hat{\mathbf{y}}^{(i)}$ and renormalize it, leading to a new probabilistic training label:

$$\forall c \in [C], \qquad \hat{\mathbf{y}}_{-j'c'}^{(i)} = \frac{\sigma(\sum_{j=1}^M \mathbb{1}\{c \neq c' \wedge j \neq j'\} \cdot \mathbf{W}_{j, \mathbf{L}_{ij}, c})}{\sum_{k=1}^C \sigma(\sum_{j=1}^M \mathbb{1}\{k \neq c' \wedge j \neq j'\} \cdot \mathbf{W}_{j, \mathbf{L}_{ij}, k})}. \tag{12}$$

Then, we perturb the training loss by moving the weight of the original loss on the $i$-th data $\hat{\ell}(\hat{\mathbf{y}}^{(i)}, \mathbf{f}_\theta(x^{(i)}))$ to a new loss term $\hat{\ell}(\hat{\mathbf{y}}_{-j'c'}^{(i)}, \mathbf{f}_\theta(x^{(i)}))$:

$$\hat{\mathcal{L}}_{\epsilon_{i,j,c}}^{wm}(\hat{\mathcal{D}}; \theta) = \frac{1}{N} \sum_{i'=1}^N \hat{\ell}(\hat{\mathbf{y}}^{(i')}, \mathbf{f}_\theta(x^{(i')})) + \epsilon_{i,j,c} \cdot \hat{\ell}(\hat{\mathbf{y}}^{(i)}, \mathbf{f}_\theta(x^{(i)})) - \epsilon_{i,j,c} \cdot \hat{\ell}(\hat{\mathbf{y}}_{-jc}^{(i)}, \mathbf{f}_\theta(x^{(i)})). \tag{13}$$

When $\epsilon_{i,j,c} = -\frac{1}{N}$, the loss term $\hat{\ell}(\hat{\mathbf{y}}^{(i)}, \mathbf{f}_\theta(x^{(i)}))$ is indeed replaced by the new loss $\hat{\ell}(\hat{\mathbf{y}}_{-j'c'}^{(i)}, \mathbf{f}_\theta(x^{(i)}))$. We again provide the formulation of corresponding influence score:

$$\bar{\phi}_{i,j,c}^{wm}(z') = -\nabla_\theta \ell\left(y', \mathbf{f}_{\theta^\star}(x')\right)^\top \mathbf{H}_{\theta^\star}^{-1} \nabla_\theta \hat{\ell}(\hat{\mathbf{y}}^{(i)} - \hat{\mathbf{y}}_{-jc}^{(i)}, \mathbf{f}_{\theta^\star}(x^{(i)})). \tag{14}$$

**Discussion.** Note that the change of loss over a holdout set $\mathcal{D}_v$ can be similarly computed and we denote it as $\bar{\phi}_{i,j,c}^{rw}(\mathcal{D}_v)$ and $\bar{\phi}_{i,j,c}^{wm}(\mathcal{D}_v)$. We omit the input variable and the superscript and use $\bar{\phi}_{i,j,c}$ to represent the proposed source-aware IF in general. The $\bar{\phi}_{i,j,c}$ corresponds to the influence of a specific perturbation on the training loss: setting the label model parameter $\mathbf{W}_{j, \mathbf{L}_{ij}, c}$ to zero for the $i$-the data so that the $(i, j, c)$-effect is removed from training. Then, the difference between reweighting and weight-moving reduces to whether renomarlization on the probabilistic labels is performed after the perturbation.

**Extension to RelatIF [3].** We additionally demonstrate how to compute the RelatIF [3] in our source-aware IF setting, Specifically, the self-influence is $\bar{\phi}_{i,j,c}^{self} = -\nabla_\theta \bar{\ell}_{i,j,c}(\theta^\star)^\top \mathbf{H}_{\theta^\star}^{-1} \nabla_\theta \bar{\ell}_{i,j,c}(\theta^\star)$. Then, the source-aware RelatIF for both $\bar{\phi}_{i,j,c}^{rw}$ and $\bar{\phi}_{i,j,c}^{wm}$ are $\frac{\bar{\phi}_{i,j,c}^{rw}}{\sqrt{\bar{\phi}_{i,j,c}^{self}}}$ and $\frac{\bar{\phi}_{i,j,c}^{wm}}{\sqrt{\bar{\phi}_{i,j,c}^{self}}}$, respectively.

### 3.3 Use cases

**Case 1: estimating influence of different components.** One major benefit of our training loss decomposition and the identity $\sigma(\cdot)$ function is that we could readily reuse the source-aware IF $\bar{\phi}_{i,j,c}$

Table 1: Calculation of influence of each component in PWS using the source-aware IF $\bar{\phi}_{i,j,c}$.

| Component of PWS pipeline | Symbol | Calculation |
|---|---|---|
| $x^{(i)}$: the $i$-th data | $\phi_{x^{(i)}}$ | $\sum_{j=1}^{M} \sum_{c=1}^{C} \bar{\phi}_{i,j,c}$ |
| $\lambda^{(j)}$: the $j$-th LF | $\phi_{\lambda^{(j)}}$ | $\sum_{i=1}^{N} \sum_{c=1}^{C} \bar{\phi}_{i,j,c}$ |
| $\mathbf{L}_{ij}$: the $j$-th LF's output on the $i$-th data | $\phi_{\mathbf{L}_{ij}}$ | $\sum_{c=1}^{C} \bar{\phi}_{i,j,c}$ |
| $\mathbf{W}_{j,k,c}$: a individual parameter of label model indexed by $(j,k,c)$ | $\phi_{\mathbf{W}_{j,k,c}}$ | $\sum_{i=1}^{N} \bar{\phi}_{i,j,c} \mathbb{1}\{L_{ij} = k\}$ |

to estimate different components of the PWS pipeline, *e.g.*, individual data or LF. This is because according to the study of group influence [23], when measuring the change in test prediction/loss, the influence is additive, *i.e.*, the influence of a set is the sum of influences of its constituent points. However, such a nice property only holds true for the reweighting method, while, in theory, the influence calculated by weight-moving does not follow due to the fact that they are not additive. We list the influence of different components of a PWS pipeline as well as how to calculate them using the fine-grain IF $\bar{\phi}_{i,j,c}$ in Table 1. Note that the 1st and 2nd row of Table 1 indeed recover the ordinary IF of a training data [22] and a LF [23], respectively.

**Case 2: improving test loss.** Another application of IF is to improve the test loss by identifying and then discarding/downweighting the harmful training data [24, 42, 38]. Similarly, the source-aware IF $\bar{\phi}_{i,j,c}$ can be employed to improve the test loss by discarding/downweighting the loss term $\bar{\ell}_{i,j,c}(\theta)$ with negative impact. Specifically, we have the following theorem for reweighting (a similar result for weight-moving can be found in Appendix E.2):

**Theorem 1.** *Discarding or downweighting the loss terms in* $\mathcal{S}_- = \{\bar{\ell}_{i,j,c}(\cdot) | i \in [N], j \in [M], c \in [C], \bar{\phi}_{i,j,c}^{rw}(\mathcal{D}_t) > 0\}$ *from training could lead to a model with lower loss over a holdout set* $\mathcal{D}_t$:

$$\mathcal{L}(\mathcal{D}_t; \theta_{\mathcal{S}_-}^\star) - \mathcal{L}(\mathcal{D}_t; \theta^\star) \approx -\frac{1}{N} \sum_{\bar{\ell}_{i,j,c}(\cdot) \in \mathcal{S}_-} \bar{\phi}_{i,j,c}^{rw}(\mathcal{D}_t) \leq 0$$

*where* $\theta_{\mathcal{S}_-}^\star$ *is the optimal model parameters obtained after the perturbation.*

In practice, we use the validation set $\mathcal{D}_v$ and $\mathcal{S}_-(\alpha) = \{\bar{\ell}_{i,j,c}(\cdot) | i \in [N], j \in [M], c \in [C], \bar{\phi}_{i,j,c}^{rw}(\mathcal{D}_v) > \alpha\}$ to tolerate the estimation error of IF where the hyperparameter $\alpha$ controls the proportion of items to be perturbed [24]. Note that this method of improving test loss relies on the training loss decomposition and $\sigma(\cdot)$ being the identity function.

### 3.4 Beyond the identity function

The aforementioned use cases rely on the additivity of influence score and the training loss decomposition, which are both consequences of $\sigma(\cdot)$ being the identity function. When $\sigma(\cdot)$ is the exponential function, we could approximate the label model $\mathbf{g}_{\mathbf{W}}(\cdot)$ by a new label model $\bar{\mathbf{g}}_{\bar{\mathbf{W}}}(\cdot)$ with parameter $\bar{\mathbf{W}}$ of the same shape but using identity function instead. Obviously, we want the new label model $\bar{\mathbf{g}}_{\bar{\mathbf{W}}}(\cdot)$ to reproduce the same probabilistic labels as those generated by $\mathbf{g}_{\mathbf{W}}(\cdot)$. This motivates us to solve the following optimization problem for $\bar{\mathbf{W}}$:

$$\bar{\mathbf{W}}^\star = \arg\min_{\bar{\mathbf{W}}} \sum_{i=1}^{N} \sum_{c=1}^{C} \left( \frac{\exp(\sum_{j=1}^{M} \mathbf{W}_{j,\mathbf{L}_{ij},c})}{\sum_{k=1}^{C} \exp(\sum_{j=1}^{M} \mathbf{W}_{j,\mathbf{L}_{ij},k})} - \frac{\sum_{j=1}^{M} \bar{\mathbf{W}}_{j,\mathbf{L}_{ij},c}}{\sum_{k=1}^{C} \sum_{j=1}^{M} \bar{\mathbf{W}}_{j,\mathbf{L}_{ij},k}} \right)^2. \tag{15}$$

Then, we can replace the label model $\mathbf{g}_{\mathbf{W}}(\cdot)$ with $\bar{\mathbf{g}}_{\bar{\mathbf{W}}^\star}(\cdot)$ everywhere. As a consequence, we are now able to unify various types of label models in a single framework and enjoy the convenience and advantages brought by the linearity of the identity function. Note that for the original label model $\mathbf{g}_{\mathbf{W}}(\cdot)$ with $\sigma(\cdot)$ being the exponential function, we can still derive the influence $\bar{\phi}_{i,j,c}$ (see appendix). We study the effect of replacing original label model with the approximated one on the end model performance in the experiment section.

# 4 Experiment

## 4.1 Setup

**Datasets.** We include the following four classification datasets in WRENCH [50], a collection of benchmarking datasets for PWS: **Census**, **Youtube**, **Yelp**, and **IMDb**. Note that the **Census** dataset is a tabular dataset while the others are textual datasets. We use the labeling functions (LFs) released by WRENCH [50]. We also include the following tabular datasets: **Mushroom** [12], **Spambase** [12], and **PhishingWebsites (PW)** [29], for which we follow the instruction in the WRENCH [50] codebase[3] to generate LFs from a decision tree learned on the labeled data. Finally, we follow [28] to derive LFs for a multiclass image classification task using the DomainNet [30] dataset, which contains 345 classes of images in 6 different domains: real, sketch, quickdraw, painting, infograph and clipart. We construct a classification task for each domain with the 5 classes containing the largest number of instances. We then train classifiers using the selected classes within the remaining five domains as LFs. We name the 6 resultant datasets **DN-real**, **DN-sketch**, **DN-quickdraw**, **DN-painting**, **DN-infograph**, and **DN-clipart** for each domain, respectively. Following [50], we use a pre-trained BERT model [10] to extract features for textual data and use ResNet-18 [17] pre-trained on ImageNet for image data.

**Model Training and Evaluation.** For label model, we focus on three commonly-used choices: Majority Voting (MV), Dawid-Skene model (DS) [9], and Snorkel [31]. The $\sigma(\cdot)$ function of the latter two label models is the exponential function and we use their approximated variants (as mentioned in Section 3.4) for all experiments. Following [23], we use logistic regression for the end model and defer the experiments on using neural networks to the appendix. We train the end model initialized as zero by gradient descent for 10,000 epochs with learning rate being 0.001, and then calculate the influence score on validation set $\mathcal{D}_v$ throughout the experiments. When using the test loss as evaluation metric, we use $\mathcal{D}_v$ to do model selection and report the test loss of the select model.

For simplicity, we use **IF (R-IF)** to represent ordinary IF (RelatIF) baseline; **RW (R-RW)** and **WM (R-WM)** are the proposed source-aware IF (RelatIF) calculated by the reweighting and weight-moving method respectively. We also use the term **SIF** (**R-SIF**) to represent the source-aware IF (RelatIF) when we do not care whether it is calculated by reweighting or weight-moving. Additional experiments and details are in Appendix G.

## 4.2 Experimental results

**Understand model behavior from multiple angles.** First and foremost, source-aware IF can be used to understand the end model behavior from multiple angles, while ordinary IF can only identify influential training data. Here, we take the **DN-real** dataset as an example and use reweighting-based source-aware IF (**RW**) to diagnose misclassified test data. For a single misclassified test data of the end model, we present the most "responsible" (1) training data, (2) LF, and (3) labeling (the vote of a LF on a training data), where being the most "responsible" means having the highest influence score on that data and likely being the primary cause of the misclassification. In Table 2, we show that for the same image misclassified by the end model trained with different label models, we could have totally different explanations and components of the PWS pipeline to blame on. Specifically, for a "*dog*" image misclassified as "*golf club*", we can see that for both the DS and Snorkel label models, the most "responsible" training image are the same but the most "responsible" LF and labeling are different: for DS the LF trained with images of painting domain is likely the primary cause for the misclassification, while for Snorkel we might blame more on the LF trained with clipart images. Without source-aware IF, we can hardly distinguish the cause of misclassification for DS and Snorkel based solely on the training data as the most "responsible" one is identical.

**Identify mislabeling of LFs.** One unique advantage of source-aware IF over ordinary IF is that it allows us to trace the influence of each LF's labeling on a specific training data ($\phi_{\mathbf{L}_{ij}} = \sum_{c=1}^{C} \bar{\phi}_{i,j,c}$, the 3rd row in Table 1), which is helpful in understanding the expertise of each individual LF and thus provides clues for practitioners to debug LFs, while ordinary IF can only estimate the influence of training data as a whole. We use $\phi_{\mathbf{L}_{ij}}$ as a scoring function of how likely the $j$-th LF mislabels

---

[3]https://github.com/JieyuZ2/wrench/tree/main/datasets/tabular_data

Table 2: The most "responsible" data, LF, and labeling, identified by source-aware IF, for the misclassification of end models (trained with difference label models) on the same test image. We can see that source-aware IF can reveal the cause of misclassification from multiple angles.

| Label Model | Misclassified Test Data | | Resp. Training Data | | | Resp. LF | Resp. Labeling | | |
|---|---|---|---|---|---|---|---|---|---|
| | Data | Pred. (Prob.) | Data | True Label | Weak Label (Prob.) | | LF | Data | Vote |
| MV | | golf club (0.55) | | dog | golf club (1.0) | quickdraw | quickdraw | | golf club |
| DS | | golf club (0.44) | | dog | golf club (0.71) | painting | painting | | golf club |
| Snorkel | | golf club (0.66) | | dog | golf club (0.94) | clipart | clipart | | golf club |

Table 3: Performance comparison results on identifying mislabeling of LFs. We report the average precision (AP) score averaged over LFs for each dataset. The larger the AP is, the better the method identifies mislabeling of LFs.

| Dataset | KNN | MV | | | | | | DS | | | | | | Snorkel | | | | | |
|---|---|---|---|---|---|---|---|---|---|---|---|---|---|---|---|---|---|---|---|
| | | LM | EM | RW | R-RW | WM | R-WM | LM | EM | RW | R-RW | WM | R-WM | LM | EM | RW | R-RW | WM | R-WM |
| Census | 0.810 | 0.809 | 0.787 | 0.858 | **0.868** | 0.834 | 0.841 | 0.787 | 0.787 | 0.786 | 0.797 | 0.796 | **0.799** | 0.787 | 0.787 | 0.800 | 0.800 | 0.801 | **0.813** |
| Mushroom | **0.975** | 0.923 | 0.828 | 0.932 | 0.923 | 0.948 | **0.949** | 0.828 | 0.828 | **0.900** | 0.755 | 0.881 | 0.865 | 0.828 | 0.828 | **0.897** | 0.837 | 0.874 | 0.876 |
| PW | 0.822 | 0.863 | 0.766 | 0.891 | **0.911** | 0.879 | 0.885 | 0.766 | 0.766 | **0.892** | 0.767 | 0.888 | 0.745 | 0.766 | 0.766 | **0.889** | 0.757 | 0.879 | 0.756 |
| Spambase | 0.782 | 0.772 | 0.738 | 0.916 | **0.927** | 0.878 | 0.879 | 0.738 | 0.738 | 0.769 | 0.774 | **0.786** | 0.775 | 0.738 | 0.738 | 0.816 | 0.753 | **0.833** | 0.750 |
| IMDb | 0.702 | 0.767 | 0.699 | 0.740 | **0.799** | 0.754 | 0.769 | 0.699 | 0.699 | **0.785** | 0.726 | 0.764 | 0.681 | 0.699 | 0.699 | **0.737** | 0.713 | 0.728 | 0.711 |
| Yelp | 0.752 | 0.792 | 0.731 | 0.888 | **0.922** | 0.830 | 0.842 | 0.731 | 0.731 | **0.844** | 0.736 | 0.784 | 0.728 | 0.731 | 0.731 | **0.907** | 0.800 | 0.785 | 0.755 |
| Youtube | 0.831 | **0.949** | 0.826 | 0.861 | 0.917 | 0.853 | 0.869 | 0.826 | 0.826 | **0.908** | 0.846 | 0.877 | 0.829 | 0.826 | 0.826 | 0.927 | **0.933** | 0.894 | 0.889 |
| DN-real | 0.711 | 0.447 | 0.417 | 0.959 | **0.986** | 0.904 | 0.942 | 0.417 | 0.417 | **0.580** | 0.462 | 0.550 | 0.449 | 0.445 | 0.417 | **0.854** | 0.671 | 0.723 | 0.476 |
| DN-sketch | 0.321 | 0.339 | 0.316 | 0.764 | **0.813** | 0.669 | 0.718 | 0.316 | 0.316 | 0.506 | **0.515** | 0.460 | 0.432 | 0.316 | 0.316 | 0.497 | **0.515** | 0.455 | 0.433 |
| DN-quickdraw | 0.362 | 0.256 | 0.255 | 0.840 | **0.879** | 0.723 | 0.762 | 0.255 | 0.255 | **0.435** | 0.383 | 0.384 | 0.301 | 0.255 | 0.255 | **0.655** | 0.376 | 0.539 | 0.293 |
| DN-painting | 0.454 | 0.416 | 0.360 | 0.847 | **0.912** | 0.756 | 0.817 | 0.360 | 0.360 | **0.614** | 0.537 | 0.524 | 0.444 | 0.360 | 0.360 | **0.715** | 0.590 | 0.630 | 0.490 |
| DN-infograph | 0.361 | 0.385 | 0.356 | 0.639 | **0.714** | 0.584 | 0.644 | 0.356 | 0.356 | **0.516** | 0.512 | 0.495 | 0.479 | 0.356 | 0.356 | 0.504 | **0.526** | 0.488 | 0.464 |
| DN-clipart | 0.437 | 0.487 | 0.434 | 0.808 | **0.880** | 0.784 | 0.835 | 0.434 | 0.434 | **0.562** | 0.553 | 0.531 | 0.520 | 0.434 | 0.434 | **0.683** | 0.608 | 0.670 | 0.581 |
| Avg. | 0.640 | 0.631 | 0.578 | 0.842 | **0.881** | 0.800 | 0.827 | 0.578 | 0.578 | **0.700** | 0.643 | 0.671 | 0.619 | 0.580 | 0.578 | **0.760** | 0.683 | 0.716 | 0.637 |

$i$-th data, *i.e.*, $\mathbf{L}_{ij} \neq y^{(i)}$, to identify the mislabeling of individual LF. We thus formulate it as a binary classification task and report the average precision (AP), a standard evaluation metric of a scoring function. We use the discrepancy between $j$-th LF's labeling on $i$-th data ($\mathbf{L}_{ij}$) and the probabilistic label output by (1) label model (**LM**), (2) end model (**EM**), and (3) a $K$-nearest neighbor classifier ($K = 10$) trained on validation set (**KNN**) as baseline scoring functions. Specifically, we calculated the discrepancy as one minus the dot product of $\mathbf{L}_{ij}$'s one-hot representation and the probabilistic label; for example, if $\mathbf{L}_{ij}$'s one-hot representation is $[0, 1, 0]^\top$ and the probabilistic label is $[0.5, 0.4, 0.3]^\top$, the discrepancy is $1 - 1 \times 0.4 = 0.6$. Note that **KNN** is a more fair and competitive baseline because it directly leverages the validation set and, as our source-aware IF, does not introduce any parametrized model in addition to what PWS has. We present the results in Table 3; we can see the superiority of source-aware IF in this task since it outperformed baselines in most cases and the averaged gains are about 9%-37% for different label models. Intriguingly, when **LM** and **EM** render unsatisfactory performance, *e.g.*, on **DomainNet**, which indicates the quality of the label or end model is not good, the source-aware IF still performed well. This means it is less sensitive to the quality of PWS, which is critical for debugging the pipeline.

**Improve the test loss of the end model.** We compared the proposed source-aware IF as well as its RelatIF variant to ordinary IF/RelatIF in terms of the capability of improving the test loss. For source-aware IF/RelatIF, we discarded loss terms in $\mathcal{S}_-(\alpha)$ and then re-trained the end model (as described in Section 3.3), while for IF/RelatIF, we likewise discarded training data in $\mathcal{D}_-(\alpha) = \{z^{(i)} | i \in [N], \phi_i(\mathcal{D}_v) > \alpha\}$. For these methods, we tuned the hyperparameter $\alpha$ on validation set for best validation loss. We additionally included the group influence (**GIF**) [23] that estimates the influence of each LF and is only applicable to **MV** as in [23]; we discarded $k$ LFs with highest negative impact and tune $k$ on validation set. The results are in Table 4, where **ERM (Emperical Risk Minimization)** is the ordinary training without any perturbation. From the results, we conclude that although ordinary IF/RelatIF can already improve the test loss over ordinary training (**ERM**) by a large margin (>0.1 test loss on average), the source-aware IF/RelatIF can further boost the performance by a similar margin (>0.1 averaged test loss improvement over IF/RelatIF). This shows

Table 4: Performance comparison results on the test loss of end models.

| | MV | | | | | | | | DS | | | | | | | Snorkel | | | | | | |
|---|---|---|---|---|---|---|---|---|---|---|---|---|---|---|---|---|---|---|---|---|---|---|
| Dataset | ERM | IF | GIF | R-IF | RW | R-RW | WM | R-WM | ERM | IF | R-IF | RW | R-RW | WM | R-WM | ERM | IF | R-IF | RW | R-RW | WM | R-WM |
| Census | 0.433 | 0.369 | 0.359 | 0.372 | **0.353** | 0.361 | 0.362 | 0.362 | 0.660 | **0.563** | 0.603 | 0.574 | 0.577 | 0.611 | 0.618 | 0.601 | 0.441 | 0.448 | 0.401 | **0.382** | 0.391 | 0.385 |
| Mushroom | 0.238 | 0.195 | 0.202 | 0.194 | **0.156** | 0.186 | 0.162 | 0.168 | 0.389 | 0.270 | 0.299 | 0.276 | **0.215** | 0.333 | 0.350 | 0.353 | 0.198 | 0.241 | 0.188 | **0.180** | 0.186 | 0.182 |
| PW | 0.416 | **0.301** | 0.328 | 0.308 | 0.312 | 0.308 | 0.305 | 0.304 | 0.499 | 0.338 | 0.338 | 0.333 | **0.329** | 0.331 | 0.331 | 0.496 | 0.350 | 0.351 | 0.306 | 0.305 | 0.307 | **0.305** |
| Spambase | 0.469 | 0.312 | 0.330 | 0.317 | **0.295** | 0.312 | 0.309 | 0.309 | 0.513 | 0.310 | 0.321 | 0.306 | 0.341 | 0.301 | **0.293** | 0.406 | 0.284 | 0.289 | **0.258** | 0.278 | 0.262 | 0.279 |
| IMDb | 0.501 | 0.501 | 0.582 | 0.501 | 0.494 | **0.488** | 0.496 | 0.492 | 0.640 | 0.628 | 0.618 | 0.618 | **0.607** | 0.626 | 0.629 | 0.584 | 0.577 | 0.583 | 0.570 | **0.561** | 0.578 | 0.571 |
| Yelp | 0.502 | 0.498 | 0.425 | 0.502 | **0.398** | 0.411 | 0.420 | 0.420 | 0.465 | 0.462 | 0.465 | 0.382 | **0.370** | 0.454 | 0.440 | 0.693 | 0.516 | 0.693 | 0.387 | **0.385** | 0.516 | 0.507 |
| Youtube | 0.351 | 0.318 | **0.278** | 0.295 | 0.284 | 0.284 | 0.301 | 0.283 | 0.332 | **0.240** | 0.272 | 0.263 | 0.261 | 0.325 | 0.338 | 0.332 | 0.320 | 0.332 | 0.264 | **0.229** | 0.296 | 0.302 |
| DN-real | 0.990 | 0.627 | 0.581 | 0.713 | 0.162 | **0.117** | 0.162 | 0.125 | 0.868 | 0.536 | 0.543 | 0.140 | 0.137 | 0.160 | **0.115** | 0.878 | 0.566 | 0.584 | 0.149 | 0.136 | 0.131 | **0.103** |
| DN-sketch | 1.270 | 1.056 | 1.262 | 1.064 | **0.785** | 0.816 | 0.827 | 0.841 | 1.389 | 1.073 | 1.139 | **0.990** | 0.999 | 1.072 | 1.070 | 1.309 | 1.069 | 1.076 | 0.915 | 0.910 | 0.915 | **0.900** |
| DN-quickdraw | 1.443 | 1.032 | 1.362 | 1.123 | 0.661 | **0.642** | 0.811 | 0.810 | 1.337 | 0.938 | 0.953 | 0.719 | **0.664** | 0.783 | 0.792 | 1.467 | 0.993 | 1.041 | 0.855 | **0.849** | 1.099 | 1.061 |
| DN-painting | 1.197 | 0.948 | 0.966 | 0.957 | 0.419 | **0.386** | 0.497 | 0.427 | 1.024 | 0.769 | 0.840 | 0.469 | 0.473 | 0.493 | 0.471 | 1.139 | 0.910 | 0.904 | 0.453 | 0.395 | 0.531 | **0.389** |
| DN-infograp | 1.270 | 1.398h | **1.191** | 1.373 | 1.288 | 1.246 | 1.247 | 1.214 | 1.208 | 1.307 | 1.371 | 1.284 | 1.303 | **1.206** | 1.243 | 1.220 | 1.324 | 1.319 | 1.211 | **1.146** | 1.236 | 1.185 |
| DN-clipart | 1.041 | 1.022 | **0.872** | 0.974 | 0.971 | 1.082 | 1.172 | 0.948 | 1.015 | 0.993 | **0.793** | 0.988 | 1.007 | 0.950 | 1.117 | 0.921 | 0.704 | 0.790 | **0.650** | 0.735 | 0.814 | 0.893 |
| Avg. | 0.779 | 0.660 | 0.672 | 0.669 | **0.506** | 0.511 | 0.544 | 0.516 | 0.795 | 0.648 | 0.658 | 0.565 | **0.560** | 0.588 | 0.601 | 0.800 | 0.635 | 0.666 | 0.508 | **0.499** | 0.559 | 0.543 |

the benefit of employing IF in a source-aware manner and provides insights for practitioners to improve the PWS pipeline.

**Discussion.** One may develop sophisticated methods to identify mislabeling of LFs or to improve test loss and achieve better performance than source-aware IF, but source-aware IF is designed as a tool of interpreting the end model prediction and when used in these applications, it does not introduce additional parameterized model. Also, we believe that source-aware IF can inspire advanced methods in the future and be leveraged by them to further boost the performance on the task of identifying mislabeling of LFs or improving test loss.

### 4.3 Additional studies

**Estimate ordinary data-level IF via source-aware IF.** As described in Section 3.3, we can use the source-aware IF calculated by the reweighting method to estimate the data-level influence score ($\phi_{x^{(i)}} = \sum_{j=1}^{M} \sum_{c=1}^{C} \bar{\phi}_{i,j,c}$, the 1st row in Table 1), which coincides with the influence computed by ordinary IF in theory. Thus, we are curious about how well the source-aware IF can be used to estimate the ordinary data-level IF. Because IF is known to be noisy in the value but good in their ranking [4, 23], we instead study the ranking correlation between $\phi_{x^{(i)}}$ and the ordinary IF. In Table 5 we report the Spearman's ranking correlation coefficient ($\leq 1$), where larger coefficients indicate better estimation of the ordinary IF based on $\phi_{x^{(i)}}$ in terms of the ranking. From the results, we can see that the $\phi_{x^{(i)}}$ actually preserves the ranking of ordinary IF quite well as Spearman's coefficients are from 0.71 to 0.99, which further demonstrates the efficacy of source-aware IF.

Table 5: Spearman's ranking correlation coefficient ($\leq 1$) between influence calculated by ordinary data-level IF and that estimated from source-aware IF. All results pass the significance test ($\rho < 0.05$).

| Dataset | Census | Mushroom | PW | Spambase | IMDb | Yelp | Youtube | DN-real | DN-sketch | DN-quickdraw | DN-painting | DN-infograph | DN-clipart | Avg. |
|---|---|---|---|---|---|---|---|---|---|---|---|---|---|---|
| MV | 0.987 | 0.981 | 0.955 | 0.989 | 0.827 | 0.898 | 0.926 | 0.890 | 0.764 | 0.877 | 0.873 | 0.760 | 0.777 | 0.885 |
| DS | 0.980 | 0.975 | 0.990 | 0.992 | 0.904 | 0.640 | 0.769 | 0.893 | 0.777 | 0.867 | 0.854 | 0.730 | 0.821 | 0.861 |
| Snorkel | 0.969 | 0.981 | 0.988 | 0.987 | 0.878 | 0.959 | 0.716 | 0.891 | 0.756 | 0.876 | 0.861 | 0.711 | 0.754 | 0.871 |

**Efficacy of label model approximation.** Since we bypassed the non-linearity of exponential function in DS and Snorkel with their approximated variants (Section 3.4), it is natural to ask how well the approximated label model can perform and whether or not the approximation is well-haved. To answer these questions, in Figure 2 we show that although the averaged test loss of ordinary training (**ERM**) and best of IF/R-IF (**Best IF/R-IF**) are slightly better than their counterpart of approximated label model (**Approx. ERM** and **Approx. Best IF/R-IF**), the best averaged test loss obtained by SIF/R-SIF with the approximated label model (**Approx. Best SIF/R-SIF**) is much lower. Such an observation indicates that with the goal of improving test loss, it is beneficial to choose $\sigma(\cdot)$ to be the identity function so that the training loss can be decomposed and perturbed based on source-aware IF.

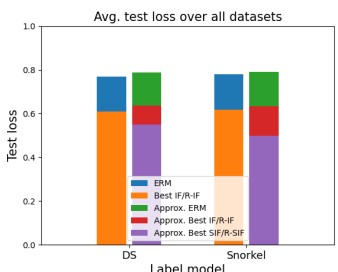

Figure 2: Avg. test loss of the end model with/without label model approximation.

# 5 Related work

**Programmatic Weak Supervision.**    In the Programmatic Weak Supervision (PWS) literature [33, 46], popular choices of labeling functions (LFs) include user-crafted heuristics [32, 39, 45], pretrained models [2, 28], external knowledge bases [18, 26], and crowd-sourced labels [9, 25]. Recently, researchers have developed various label models [33, 31, 14, 40, 48, 44, 37, 47, 47, 49, 43]. In the PWS pipeline, the behavior of the end model is directly dictated by the soft labels, wherein the soft labels themselves are also intermediate products of the label model and the LFs' votes. Thus, to interpret the trained end model and potentially debug the PWS workflow, it is critical to understand the influence of each PWS component on the end model's performance, which has yet received adequate research attention.

**Influence Function.**    Among various ways to understand a model's behavior through the training data points, Influence Function (IF) is an effective technique with a wide range of applications [22, 23, 16, 7]. In PWS, although one can directly apply IF to understand a model's behavior through the weakly labeled training data, more insights may be unveiled by further investigations of how each PWS component, e.g., LFs and label models, influences the end model. In similar spirit to us, [6] developed multi-stage IF to trace a fine-tuned model's behavior back to the pretraining data. Perhaps most similarly to our work, [23] conducted initial exploration on using group influence to study how removing an LF from the PWS pipeline would affect the end model's performance. However, the group influence method is only applicable to a specific simplified setting of PWS.

# 6 Limitation and social impacts

**Limitations.**    Although we study several popular label models and show how to use approximated label model in case that users need to use a label model that is not included in this work, user may still encounter difficulties in real applications when using complicated label model. Also, this work focuses on the two-stage PWS pipeline, while very recently there are some one-stage methods, to which our framework may not be applicable. In the main body of the paper, we use logistic regression as end model in our experiments, while in appendix we present experimental results of a simple two-layer neural network. But we do not include experiments on complicated deep learning models like Convolutional Neural Network.

**Social impacts.**    This work aims at help people understanding the Programmatic Weak Supervision and has the potential to resolve the bias in the generation process of training label, which might have positive social impact. We do not foresee any form of negative social impact induced by our work.

# 7 Conclusion

In this work, by leveraging the knowledge of how probabilistic training labels are aggregated from different sources in the PWS pipeline, we have proposed source-aware IF that enables users to estimate the influence of different components (*e.g.*, source vote, training data, LFs, *etc.*.) in the pipeline on the end model's behavior, and unlocked various practical use cases. Specifically, we have demonstrated how the proposed source-aware IF can be used to (1) unveil the most influential training data point, LF, or a specific vote to a misclassification made by a model, (2) identify mislabeling of LFs on certain data points, and (3) improve the end model's test performance.

# 8 Acknowledgement

Thanks the anonymous reviewers for their helpful comments and suggestions. We also thank Haojie Jia for his help of experiments.

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
