# A Label model reparametarization and illustrations

## A.1 Majority Voting

The Majority Voting (MV) is the most intuitive algorithm for aggregate LFs' annotations. The MV methods can be formalize as

$$\hat{\mathbf{y}}_c^{(i)} = \frac{\sum_{j=1}^M \mathbb{1}\{\mathbf{L}_{ij} = c\}}{\sum_{k=1}^C \sum_{j=1}^M \mathbb{1}\{\mathbf{L}_{ij} = k\}} \tag{16}$$

By letting $\mathbf{W}_{j,k,c} = \mathbb{1}\{k = c\}$, we re-formulate the above as

$$\forall c \in [C], \qquad \hat{\mathbf{y}}_c^{(i)} = \frac{\sum_{j=1}^M \mathbf{W}_{j,\mathbf{L}_{ij},c}}{\sum_{k=1}^C \sum_{j=1}^M \mathbf{W}_{j,\mathbf{L}_{ij},k}}. \tag{17}$$

## A.2 Dawid-Skene model

The parameters of the Dawid-Skene(DS) [9] model are given by

$$\pi_{c,l}^{(j)} = \frac{\text{number of times the } j\text{-th LF votes for } l \text{ when true label is } c}{\text{number of data voted by } j\text{-th LF whose true label is } c}, \tag{18}$$

and $p_c$, which is prior of $p(y = c)$. Let $n_{i,l}^{(j)}$ be the number of times $j$-th LF assigns $l$ for data $x^{(i)}$ and $T_{i,c}$ be the indicator variables: if the true label of data $x^{(i)}$ is $c$, then $T_{i,c} = 1$, otherwise 0. When the true label for all data are available, the likelihood is given by

$$\prod_{i=1}^N \prod_{c=1}^C \left\{ p_c \prod_{j=1}^M \prod_{l=1}^{C+1} \left( \pi_{c,l}^{(j)} \right)^{n_{i,l}^{(j)}} \right\}^{T_{i,c}} \tag{19}$$

By the Bayes' theorem:

$$\hat{\mathbf{y}}_c^{(i)} = p(y^{(i)} = c \mid \mathbf{L}_i) = p(T_{i,c} = 1 \mid \mathbf{L}_i) = \frac{\prod_{j=1}^M \prod_{l=1}^{C+1} p_c \left( \pi_{c,l}^{(j)} \right)^{n_{i,l}^{(j)}}}{\sum_{k=1}^C \left( \prod_{j=1}^M \prod_{l=1}^{C+1} p_k \left( \pi_{k,l}^{(j)} \right)^{n_{i,l}^{(j)}} \right)}. \tag{20}$$

We can re-write Eq. 21 as

$$\hat{\mathbf{y}}_c^{(i)} = \frac{\sigma_{exp} \left( \sum_{j=1}^M \sum_{l=1}^{C+1} \left( n_{i,l}^{(j)} \log \pi_{c,l}^{(j)} \right) + \log p_c \right)}{\sum_{k=1}^C \left( \sigma_{exp} \left( \sum_{j=1}^M \sum_{l=1}^{C+1} \left( n_{i,l}^{(j)} \log \pi_{k,l}^{(j)} \right) + \log p_k \right) \right)}, \tag{21}$$

By omitting the last $\log p_c$ term and letting $\mathbf{W}_{j,\mathbf{L}_{ij},c} = \sum_{l=1}^{C+1} \left( n_{i,l}^{(j)} \log \pi_{c,l}^{(j)} \right)$, we have

$$\forall c \in [C], \quad \hat{\mathbf{y}}_c^{(i)} = \frac{\sigma_{exp}(\sum_{j=1}^M \mathbf{W}_{j,\mathbf{L}_{ij},c})}{\sum_{k=1}^C \sigma_{exp}(\sum_{j=1}^M \mathbf{W}_{j,\mathbf{L}_{ij},k})}. \tag{22}$$

The additional $\log p_c$ term can be absorbed in the summation by introducing an additional LF whose label space consists of only one element and the corresponding parameter in $\mathbf{W}$ is $\mathbf{p} = [\log(p(y=1)), \log(p(y=2)), \cdots, \log(p(y=C))]^\top$. We omit this case for simplicity.

## A.3 Snorkel MeTaL

The parameters $\mu$ of Snorkel MeTaL [31] are given by

$$\mu_{j,c,m^{(j)}} = p(y = c, \lambda^{(j)} = m^{(j)}) = \mathbb{E}\left[ T_{j,c,m^{(j)}} \right], \tag{23}$$

where $T_{j,c,m^{(j)}}$ is the indicator, $T_{j,c,m^{(j)}} = \mathbb{1}\{y = c, \lambda^{(j)} = m^{(j)}\}$. Given the prior of $p(y)$, by the Bayes' theorem we have:

$$p_\mu(y = c, \boldsymbol{\lambda} = \boldsymbol{m}) = p_\mu(\boldsymbol{\lambda} = \boldsymbol{m} \mid y = c)p(y = c) = \frac{\prod_{j=1}^M p_\mu(y = c, \lambda^{(j)} = m^{(j)})}{p(y = c)^{M-1}}. \tag{24}$$

We can further infer that

$$\begin{aligned}
p_\mu(y = c \mid \boldsymbol{\lambda} = \boldsymbol{m}) &= \frac{p_\mu(y = c, \boldsymbol{\lambda} = \boldsymbol{m})}{\sum_{k=1}^C p_\mu(y = k, \boldsymbol{\lambda} = \boldsymbol{m})} \\
&= \frac{\prod_{j=1}^M p_\mu(y = c, \lambda^{(j)} = m^{(j)})}{p(y = c)^{M-1}} \Big/ \sum_{k=1}^C \frac{\prod_{j=1}^M p_\mu(y = k, \lambda^{(j)} = m^{(j)})}{p(y = k)^{M-1}} \\
&= \frac{p_c \prod_{j=1}^M (\frac{\mu_{j,c,m^{(j)}}}{p_c})}{\sum_{k=1}^C p_k \prod_{j=1}^M (\frac{\mu_{j,k,m^{(j)}}}{p_k})}.
\end{aligned} \tag{25}$$

By letting $\mathbf{W}_{j,k,c} = \log \frac{\mu_{j,c,k}}{p_c}$ and, again, omitting the $\log p_c$ term similar to the above derivation for the DS model, we re-formulate the above as

$$\forall c \in [C], \qquad \hat{\mathbf{y}}_c^{(i)} = \frac{\sigma_{exp}\left(\sum_{j=1}^M \mathbf{W}_{j,\mathbf{L}_{ij},c}\right)}{\sum_{k=1}^C \sigma_{exp}\left(\sum_{j=1}^M \mathbf{W}_{j,\mathbf{L}_{ij},k}\right)}. \tag{26}$$

### A.4 More complicated label model

For more complicated label models that we cannot formulate the inferred label as Eq. 6, we can still use the approximation described in Section 3.4. Consider a label model $\mathbf{g}(\mathbf{L}(x), x) \in \mathcal{F}$ in arbitrary functional class $\mathcal{F}$, *e.g.*, neural network, and having additional dependency on data feature $x$[4], we can still approximate such complicated function with identity function-based label model $\bar{\mathbf{g}}_{\bar{\mathbf{W}}(x)}(\mathbf{L}(x))$ similar to the aforementioned one except that $\bar{\mathbf{W}}(x) : \mathcal{X} \to \mathbb{R}^{M \times (C+1) \times C}$ is a similarly complicated function, *e.g.*, neural network, that maps each data $x \in \mathcal{X}$ to a unique label model parameter $\bar{\mathbf{W}}(x)$. We leave the exploration of more complicated form of label models into future work.

## B Influence Function derivation: the reweighting method

Fellow the derivation of [22], we can directly compute the influence of fine-grained sample $(x^{(i)}, \frac{1}{s_i}\mathbf{W}_{j,\mathbf{L}_{ij},c})$ through up-weighting its corresponding fine-grained level loss with $\epsilon_{i,j,c}$. In the following, we explicitly define the reweighted loss function with different $\sigma(\cdot)$ function.

### B.1 Case 1: identity function

We define the loss with reweighted sample as,

$$\hat{\mathcal{L}}_{\epsilon_{i,j,c}}(\hat{\mathcal{D}}; \theta) = \frac{1}{N} \sum_{i'=1}^N \sum_{j'=1}^M \sum_{c'=1}^C \bar{\ell}_{i',j',c'}(\theta) + \epsilon_{i,j,c} \cdot \bar{\ell}_{i,j,c}(\theta), \tag{27}$$

and the corresponding risk minimizer as $\theta_{\epsilon_{i,j,c}}^\star$. According to the Equation (4), the influence on the loss of one test sample $(x_k, y_k) \in \mathcal{D}_v$ is $-\nabla_\theta \hat{\ell}(\mathbf{y}_k, \mathbf{f}_{\theta^\star}(x_k))^\top \mathbf{H}_{\theta^\star}^{-1} \nabla_\theta \bar{\ell}_{i,j,c}(\theta^\star) \cdot \epsilon_{i,j,c}$, where $\mathbf{H}_{\theta^\star}^{-1} = \frac{1}{N} \sum_{i,j,c} \nabla_\theta^2 \bar{\ell}_{i,j,c}(\theta^\star)$.

The influence score on the loss of $z' = (x', y')$ by discarding the loss term $\bar{\ell}_{i,j,c}(\theta)$ is,

$$\bar{\phi}_{i,j,c}^{rw}(z') = \nabla_\theta \hat{\ell}(\mathbf{y}_k, \mathbf{f}_{\theta^\star}(x_k))^\top \mathbf{H}_{\theta^\star}^{-1} \nabla_\theta \bar{\ell}_{i,j,c}(\theta^\star). \tag{28}$$

And the influence on the loss on the set $\mathcal{D}_v$ can be defined as,

$$\bar{\phi}_{i,j,c}^{rw}(\mathcal{D}_v) = -\frac{1}{|\mathcal{D}_v|} \sum_{z' \in \mathcal{D}_v} \bar{\phi}_{i,j,c}^{rw}(z'). \tag{29}$$

---

[4]Several recent neural network-based label models roughly follow this functional form [34, 21].

## B.2 Case 2: exponential function

With exponential function, the label for sample $x^{(i)}$ can be presented as,

$$\hat{\mathbf{y}}_c^{(i)} = \frac{\exp(\sum_{j=1}^M \mathbf{W}_{j,\mathbf{L}_{ij},c})}{\sum_{k=1}^C \exp(\sum_{j=1}^M \mathbf{W}_{j,\mathbf{L}_{ij},k})},$$

where $c \in [C]$. For the sample $x^{(i)}$, if the value of $j$-th label function on $c$-th class is upweighted with $\epsilon_{i,j,c}$, then the corresponding label can be written as,

$$\hat{\mathbf{y}}_{c,\epsilon_{i,j,c}}^{(i)} = \frac{\exp(\sum_{j'=1}^M \mathbf{W}_{j',\mathbf{L}_{ij'},c} + \epsilon_{i,j,c}\mathbf{W}_{j,\mathbf{L}_{ij},c})}{\sum_{k'=1}^C \exp(\sum_{j'=1}^M \mathbf{W}_{j',\mathbf{L}_{ij'},k'} + \epsilon_{i,j,c}\mathbf{W}_{j,\mathbf{L}_{ij},c})}$$

$$= \frac{\exp(\sum_{j'=1}^M \mathbf{W}_{j',\mathbf{L}_{ij'},c} + \epsilon_{i,j,c}\mathbf{W}_{j,\mathbf{L}_{ij},c})}{\sum_{k'=1}^C \exp(\sum_{j'=1}^M \mathbf{W}_{j',\mathbf{L}_{ij'},k'}) + \exp(\mathbf{W}_{j,\mathbf{L}_{ij},c}) \cdot (\exp(\epsilon_{i,j,c}\mathbf{W}_{j,\mathbf{L}_{ij},c}) - 1)}.$$

We assume that $\sum_{k'=1}^C \exp(\sum_{j'=1}^M \mathbf{W}_{j',\mathbf{L}_{ij'},k'}) \gg \exp(\mathbf{W}_{j,\mathbf{L}_{ij},c}) \cdot (\exp(\epsilon_{i,j,c}\mathbf{W}_{j,\mathbf{L}_{ij},c}) - 1)$. Note, this assumption holds in general, because $\epsilon_{i,j,c}$ is close to zero as we discussed in the Section 2. Then we have,

$$\hat{\mathbf{y}}_{c,\epsilon_{i,j,c}}^{(i)} = \frac{\exp(\sum_{j'=1}^M \mathbf{W}_{j',\mathbf{L}_{ij'},c} + \epsilon_{i,j,c}\mathbf{W}_{j,\mathbf{L}_{ij},c})}{\sum_{k'=1}^C \exp(\sum_{j'=1}^M \mathbf{W}_{j',\mathbf{L}_{ij'},k'})}$$

$$= \hat{\mathbf{y}}_c^{(i)} + \exp(\epsilon_{i,j,c}\mathbf{W}_{j,\mathbf{L}_{ij},c} - 1) \cdot \hat{\mathbf{y}}_c^{(i)}.$$

Then, the reweighted risk over the whole training set becomes

$$\hat{\mathcal{L}}_{\epsilon_{i,j,c}}(\hat{\mathcal{D}}; \theta) = -\frac{1}{N} \left( \sum_{i'=1}^N \sum_{c'=1}^C \hat{\mathbf{y}}_{c'}^{(i')} \log\left(\mathbf{f}_\theta(x^{(i')})_{c'}\right) + \exp(\epsilon_{i,j,c}\mathbf{W}_{j,\mathbf{L}_{ij},c} - 1) \cdot \hat{\mathbf{y}}_c^{(i)} \log\left(\mathbf{f}_\theta(x^{(i)})_c\right) \right). \tag{30}$$

We denote the minimizer of the reweighted risk $\hat{\mathcal{L}}_{\epsilon_{i,j,c}}(\hat{\mathcal{D}}; \theta)$ as $\theta_{\epsilon_{i,j,c}}^{rw^\star}$. Then the change of parameters can be presented as,

$$\theta_{\epsilon_{i,j,c}}^{rw^\star} - \theta^\star = -\mathbf{H}_{\theta^\star}^{-1} \nabla_\theta\left(\hat{\mathbf{y}}_c^{(i)} \log\left(\mathbf{f}_{\theta^\star}(x^{(i)})_c\right)\right) \cdot \exp(\epsilon_{i,j,c}\mathbf{W}_{j,\mathbf{L}_{ij},c} - 1). \tag{31}$$

And the change with respect to $\epsilon_{i,j,c}$ is,

$$\frac{d(\theta_{\epsilon_{i,j,c}}^{rw^\star} - \theta^\star)}{d\epsilon_{i,j,c}} = \frac{d\theta_{\epsilon_{i,j,c}}^{rw^\star}}{d\epsilon_{i,j,c}} = -\mathbf{H}_{\theta^\star}^{-1} \nabla_\theta\left(\hat{\mathbf{y}}_c^{(i)} \log\left(\mathbf{f}_{\theta^\star}(x^{(i)})_c\right)\right) \cdot \frac{d\exp(\epsilon_{i,j,c}\mathbf{W}_{j,\mathbf{L}_{ij},c} - 1)}{d\epsilon_{i,j,c}} \tag{32}$$

According to the Talyor expansion, we have $\exp(x) = 1 + x + O(x)$. Then, we can obtain,

$$\frac{d(\theta_{\epsilon_{i,j,c}}^{rw^\star} - \theta^\star)}{d\epsilon_{i,j,c}} = \frac{d\theta_{\epsilon_{i,j,c}}^{rw^\star}}{d\epsilon_{i,j,c}} = -\mathbf{H}_{\theta^\star}^{-1} \nabla_\theta\left(\hat{\mathbf{y}}_c^{(i)} \log\left(\mathbf{f}_{\theta^\star}(x^{(i)})_c\right)\right) \cdot \mathbf{W}_{j,\mathbf{L}_{ij},c}. \tag{33}$$

The influence score on the loss of $z' = (x', y')$ by discarding the loss term $\bar{\ell}_{i,j,c}(\theta)$ is,

$$\bar{\phi}_{i,j,c}^{rw}(z') = \nabla_\theta\hat{\ell}\left(\mathbf{y}_k, \mathbf{f}_{\theta^\star}(x_k)\right)^\top \mathbf{H}_{\theta^\star}^{-1} \nabla_\theta\left(\hat{\mathbf{y}}_c^{(i)} \log\left(\mathbf{f}_{\theta^\star}(x^{(i)})_c\right)\right) \cdot \mathbf{W}_{j,\mathbf{L}_{ij},c}. \tag{34}$$

## C  Influence Function derivation: the weight-moving method

Instead of employing the decomposing loss function, we introduce a more general influence estimation method - weight-moving Influence, which get ride of the loss decomposition and approximation and is agnostic to the selection of $\sigma(\cdot)$ function. As we introduced previously, the label of sample $x^{(i)}$ for each class $c$ can be defined as:

$$\forall c \in [C], \qquad \hat{\mathbf{y}}_c^{(i)} = \frac{\sigma(\sum_{j=1}^M \mathbf{W}_{j,\mathbf{L}_{ij},c})}{\sum_{k=1}^C \sigma(\sum_{j=1}^M \mathbf{W}_{j,\mathbf{L}_{ij},k})}. \tag{35}$$

Further, we define the label after removing the output value of $j'$-th label function on the $c'$-th class for sample $i$ as:

$$\forall c \in [C], \qquad \hat{\mathbf{y}}^{(i)}_{-j'c',c} = \frac{\sigma(\sum_{j=1}^{M} \mathbb{1}[c \neq c' \vee j \neq j'] \cdot \mathbf{W}_{j,\mathbf{L}_{ij},c})}{\sum_{k=1}^{C} \sigma(\sum_{j=1}^{M} \mathbb{1}[k \neq c' \wedge j \neq j'] \cdot \mathbf{W}_{j,\mathbf{L}_{ij},k})}. \tag{36}$$

Similarly, we can also define the label vector $\hat{\mathbf{y}}^{(i)}_{-j'c'}$ for the sample $x^{(i)}$. Then, we define the minimizer of the weight-moving loss as:

$$\theta^{wm\star}_{\epsilon_{i,j,c}} = \arg\min \frac{1}{N}\sum_{i'=1}^{N} \hat{\ell}(\hat{\mathbf{y}}^{(i')}, \mathbf{f}_\theta(x^{(i')})) + \epsilon_{i,j,c} \cdot \hat{\ell}(\hat{\mathbf{y}}^{(i)}, \mathbf{f}_\theta(x^{(i)})) - \epsilon_{i,j,c} \cdot \hat{\ell}(\hat{\mathbf{y}}^{(i)}_{-jc}, \mathbf{f}_\theta(x^{(i)})). \tag{37}$$

The change of parameters with respect to $\epsilon_{i,j,c}$ can be written as,

$$\frac{d(\theta^{wm\star}_{\epsilon_{ijc}} - \theta^\star)}{d\epsilon_{i,j,c}} = \frac{d\theta^{wm\star}_{\epsilon_{ijc}}}{d\epsilon_{i,j,c}} \tag{38}$$

$$= -\mathbf{H}_{\theta^\star}^{-1}\Big[\nabla_\theta \sum_{c'=1}^{C} \hat{\mathbf{y}}^{(i)}_{c'} \log\left(\mathbf{f}_{\theta^\star}(x^{(i)})\right) - \nabla_\theta \sum_{c'=1}^{C} \hat{\mathbf{y}}^{(i)}_{-jc,c'} \log\left(\mathbf{f}_{\theta^\star}(x^{(i)})\right)\Big] \tag{39}$$

$$= -\mathbf{H}_{\theta^\star}^{-1}\nabla_\theta \hat{\ell}(\hat{\mathbf{y}}^{(i)} - \hat{\mathbf{y}}^{(i)}_{-jc}, \mathbf{f}_{\theta^\star}(x^{(i)})) \tag{40}$$

With $\epsilon_{i,j,c} = -\frac{1}{N}$, the weight for the sample $(x^{(i)}, \hat{\mathbf{y}}^{(i)})$ is moved to the the sample $(x^{(i)}, \hat{\mathbf{y}}^{(i)}_{-jc})$. Then, the influence of the loss on sample $z' = (x', y')$ by removing the output value of $j$-th label function on the $c$-th class for sample $i$ is,

$$\bar{\phi}^{wm}_{i,j,c}(z') = -\nabla_\theta \ell\left(y', \mathbf{f}_{\theta^\star}(x')\right)^\top \mathbf{H}_{\theta^\star}^{-1}\nabla_\theta \hat{\ell}(\hat{\mathbf{y}}^{(i)} - \hat{\mathbf{y}}^{(i)}_{-jc}, \mathbf{f}_{\theta^\star}(x^{(i)})). \tag{41}$$

We then define the weight-moving influence over the set $\mathcal{D}_v$ as,

$$\bar{\phi}^{wm}_{i,j,c}(\mathcal{D}_v) = \frac{1}{|\mathcal{D}_v|}\sum_{z'\in\mathcal{D}_v} \bar{\phi}^{wm}_{i,j,c}(z') \tag{42}$$

## D   Connection between weight-moving and reweighting method

To show the connection between weight-moving influence ($\bar{\phi}^{wm}_{i,j,c}$) and reweighting influence ($\bar{\phi}^{rw}_{i,j,c}$), we firstly decompose the weight-moving loss (37). For the identity function, we have:

$$\mathcal{L}^{wm} = \frac{1}{N}\sum_{i'=1}^{N} \hat{\ell}(\hat{\mathbf{y}}^{(i')}, \mathbf{f}_\theta(x^{(i')})) + \epsilon_{i,j,c} \cdot \hat{\ell}(\hat{\mathbf{y}}^{(i)}, \mathbf{f}_\theta(x^{(i)})) - \epsilon_{i,j,c} \cdot \hat{\ell}(\hat{\mathbf{y}}^{(i)}_{-jc}, \mathbf{f}_\theta(x^{(i)})) \tag{43}$$

$$= \frac{1}{N}\sum_{i'=1}^{N} \hat{\ell}(\hat{\mathbf{y}}^{(i')}, \mathbf{f}_\theta(x^{(i')})) + \epsilon_{i,j,c}\Big[-\sum_{c'=1}^{C}\frac{\sum_{j=1}^{M}\mathbf{W}_{j,\mathbf{L}_{ij},c'}}{\sum_{k=1}^{C}\sum_{j=1}^{M}\mathbf{W}_{j,\mathbf{L}_{ij},k}}\log\left(\mathbf{f}_\theta(x^{(i)})_{c'}\right)\Big] \tag{44}$$

$$- \epsilon_{i,j,c}\Big[-\sum_{c'=1}^{C}\frac{\sum_{j'=1}^{M}\mathbb{1}[c \neq c' \vee j \neq j']\cdot\mathbf{W}_{j',\mathbf{L}_{ij'},c'}}{\sum_{k=1}^{C}\sum_{j'=1}^{M}\mathbb{1}[k \neq c' \wedge j \neq j']\cdot\mathbf{W}_{j',\mathbf{L}_{ij'},k}}\cdot\log\left(\mathbf{f}_\theta(x^{(i)})_{c'}\right)\Big] \tag{45}$$

For simplicity, we denote $C_i = \sum_{k=1}^{C}\sum_{j=1}^{M}\mathbf{W}_{j,\mathbf{L}_{ij},k}$ and $C'_i = \sum_{k=1}^{C}\sum_{j'=1}^{M}\mathbb{1}\{k \neq c', j \neq j'\}\cdot\mathbf{W}_{j',\mathbf{L}_{ij'},k}$. Note $\frac{C'_i}{C_i} \to 1$, with $j \to \infty$ or $C \to \infty$. Further, we assume $\frac{C'_i}{C_i} \approx 1$. Then, we have,

$$\mathcal{L}^{wm} = \frac{1}{N}\sum_{i'=1}^{N} \hat{\ell}(\hat{\mathbf{y}}^{(i')}, \mathbf{f}_\theta(x^{(i')})) + \epsilon_{i,j,c}\Big[-\frac{\mathbf{W}_{j,\mathbf{L}_{ij},c}}{C_i}\log\left(\mathbf{f}_\theta(x^{(i)})_c\right)\Big] \tag{46}$$

$$\approx \frac{1}{N}\sum_{i'=1}^{N} \hat{\ell}(\hat{\mathbf{y}}^{(i')}, \mathbf{f}_\theta(x^{(i')})) + \epsilon_{i,j,c} \cdot \bar{\ell}_{i,j,c}(\theta). \tag{47}$$

Note, the above loss function is same as the loss we defined in the Equation (27). Therefore, the weight-moving influence, $\bar{\phi}^{wm}_{i,j,c}(\mathcal{D}_v)$, is an approximation of the reweighting influence, $\bar{\phi}^{rw}_{i,j,c}(\mathcal{D}_v)$, especially in the setting with large number of labeling function $M$ and large number of class $C$.

# E   Proofs for theoretical analysis

We first show the correctness of Theorem 1, in which we employ the reweighting influence as a proxy to downweight or discard data samples. Then, we show that the similar conclusion can be obtained with weight-moving influence as the proxy. We summarized the conclusion for weight-moving influence into the Theorem 2.

## E.1   Proofs for Theorem 1

**Theorem 1.** *Discarding or downweighting the loss terms in* $\mathcal{S}_- = \{\bar{\ell}_{i,j,c}(\cdot)|i \in [N], j \in [M], c \in [C], \bar{\phi}_{i,j,c}^{rw}(\mathcal{D}_t) > 0\}$ *from training could lead to a model with lower loss over a holdout set* $\mathcal{D}_t$:

$$\mathcal{L}(\mathcal{D}_t; \theta_{\mathcal{S}_-}^\star) - \mathcal{L}(\mathcal{D}_t; \theta^\star) \approx -\frac{1}{N} \sum_{\bar{\ell}_{i,j,c}(\cdot) \in \mathcal{S}_-} \bar{\phi}_{i,j,c}^{rw}(\mathcal{D}_t) \leq 0$$

*where* $\theta_{\mathcal{S}_-}^\star$ *is the optimal model parameters obtained after the perturbation.*

As we derived in Appendix B.1, the reweighting influence represent that the change of loss over the set $\mathcal{D}_t$ through upweighting the fine-grained sample $(x^{(i)}, \frac{1}{s_i}\mathbf{W}_{j,\mathbf{L}_{ij},c})$ by $\epsilon_{i,j,c}$,

$$\bar{\phi}_{i,j,c}^{rw}(\mathcal{D}_v) = \mathcal{L}(\mathcal{D}_t; \theta_{\mathcal{S}_-}^\star) - \mathcal{L}(\mathcal{D}_t; \theta^\star) = -\sum_{(x_k, y_k) \in \mathcal{D}_t} \nabla_\theta \hat{\ell}\left(\mathbf{y}_k, \mathbf{f}_{\theta^\star}(x_k)\right)^\top \mathbf{H}_{\theta^\star}^{-1} \nabla_\theta \bar{\ell}_{i,j,c}(\theta^\star).$$

Through downweighting or discarding (note, discarding a sample is equal to downweighting it with $\epsilon_{i,j,c} = -1/N$) the fine-grained samples with $\bar{\phi}_{i,j,c}^{rw}(\mathcal{D}_v) > 0$, the change of the loss is equal to $\epsilon_{i,j,c} \cdot \bar{\phi}_{i,j,c}^{rw} < 0$. As commonly assumed in the previous works [42, 24], the influence caused by reweighting different samples is independent. Then, with $\epsilon_{i,j,c} = -1/N$, we can get the conclusion that $\mathcal{L}(\mathcal{D}_t; \theta_{\mathcal{S}_-}^\star) - \mathcal{L}(\mathcal{D}_t; \theta^\star) \approx -\frac{1}{N} \sum_{\bar{\ell}_{i,j,c}(\cdot) \in \mathcal{S}_-} \bar{\phi}_{i,j,c}^{rw}(\mathcal{D}_t) \leq 0$.

## E.2   Proofs for Theorem 2

**Theorem 2.** *Discarding or downweighting the training loss term in* $\mathcal{S}_- = \{\bar{\ell}_{i,j,c}(\theta)|i \in [N], j \in [M], c \in [C], \bar{\phi}_{i,j,c}^{wm}(\mathcal{D}_v) > \alpha\}$ *from training could lead to a model with lower loss over a test set* $\mathcal{D}_t$:

$$\mathcal{L}(\mathcal{D}_t; \theta_{\mathcal{S}_-}^\star) - \mathcal{L}(\mathcal{D}_t; \theta^\star) \leq 0$$

*where* $\alpha \in \mathbb{R}^+$ *is a positive number close to 0, and* $\theta_{\mathcal{S}_-}^\star$ *is the optimal model parameters obtained after the perturbation.*

We first decompose the weight-moving loss in the following form,

$$
\begin{aligned}
\mathcal{L}^{wm} &= \frac{1}{N} \sum_{i'=1}^{N} \hat{\ell}(\hat{\mathbf{y}}^{(i')}, \mathbf{f}_\theta(x^{(i')})) + \epsilon_{i,j,c} \cdot \hat{\ell}(\hat{\mathbf{y}}^{(i)}, \mathbf{f}_\theta(x^{(i)})) - \epsilon_{i,j,c} \cdot \hat{\ell}(\hat{\mathbf{y}}_{-jc}^{(i)}, \mathbf{f}_\theta(x^{(i)})) \\
&= \frac{1}{N} \sum_{i'=1}^{N} \hat{\ell}(\hat{\mathbf{y}}^{(i')}, \mathbf{f}_\theta(x^{(i')})) + \epsilon_{i,j,c}\left[-\sum_{c'=1}^{C} \frac{\sum_{j=1}^{M} \mathbf{W}_{j,\mathbf{L}_{ij},c'}}{\sum_{k=1}^{C} \sum_{j=1}^{M} \mathbf{W}_{j,\mathbf{L}_{ij},k}} \log\left(\mathbf{f}_\theta(x^{(i)})_{c'}\right)\right] \\
&\quad - \epsilon_{i,j,c}\left[-\sum_{c'=1}^{C} \frac{\sum_{j'=1}^{M} \mathbb{1}[c \neq c' \vee j \neq j'] \cdot \mathbf{W}_{j',\mathbf{L}_{ij'},c'}}{\sum_{k=1}^{C} \sum_{j'=1}^{M} \mathbb{1}[k \neq c' \wedge j \neq j'] \cdot \mathbf{W}_{j',\mathbf{L}_{ij'},k}} \cdot \log\left(\mathbf{f}_\theta(x^{(i)})_{c'}\right)\right] \\
&= \frac{1}{N} \sum_{i'=1}^{N} \hat{\ell}(\hat{\mathbf{y}}^{(i')}, \mathbf{f}_\theta(x^{(i')})) + \epsilon_{i,j,c}\left[-\frac{\mathbf{W}_{j,\mathbf{L}_{ij},c}}{C_i} \log\left(\mathbf{f}_\theta(x^{(i)})_c\right)\right] \\
&\quad - \epsilon_{i,j,c}\left(\frac{1}{c_i'} - \frac{1}{c_i}\right) \sum_{j'=1}^{M} \sum_{c'=1}^{C} \mathbb{1}[c \neq c' \vee j \neq j']\left(-\mathbf{W}_{j',\mathbf{L}_{ij'},c'} \log\left(\mathbf{f}_\theta(x^{(i)})_{c'}\right)\right) \\
&= \frac{1}{N} \sum_{i'=1}^{N} \hat{\ell}(\hat{\mathbf{y}}^{(i')}, \mathbf{f}_\theta(x^{(i')})) + \epsilon_{i,j,c} \cdot \bar{\ell}_{i,j,c}(\theta) - \epsilon_{i,j,c} \cdot \tilde{L}_{i,j,c}
\end{aligned}
$$

where, we denote $\tilde{L}_{i,j,c} = (\frac{1}{c_i'} - \frac{1}{c_i}) \sum_{j'=1}^{M} \sum_{c'=1}^{C} \mathbb{1}[c \neq c' \vee j \neq j'] \left( -\mathbf{W}_{j',\mathbf{L}_{ij},c'} \log \left( \mathbf{f}_\theta(x^{(i)})_{c'} \right) \right)$ and $C_i = \sum_{k=1}^{C} \sum_{j=1}^{M} \mathbf{W}_{j,\mathbf{L}_{ij},k}$ and $C_i' = \sum_{k=1}^{C} \sum_{j'=1}^{M} \mathbb{1}[k \neq c', j \neq j'] \cdot \mathbf{W}_{j',\mathbf{L}_{ij'},k}$. Note, as we discussed in the Appendix D, $\frac{1}{c_i'} - \frac{1}{c_i}$ is a small positive value and close to 0. Therefore, $\tilde{L}_{i,j,c} \ll \bar{\ell}_{i,j,c}$.

$$\bar{\phi}_{i,j,c}^{wm}(\mathcal{D}_v) = \mathcal{L}^{wm}(\mathcal{D}_t; \theta_{\epsilon_{i,j,c}}^\star) - \mathcal{L}^{wm}(\mathcal{D}_t; \theta^\star)$$
$$= \bar{\phi}_{i,j,c}^{rw}(\mathcal{D}_v) + \sum_{(x_k,y_k) \in \mathcal{D}_t} \nabla_\theta \hat{\ell}(\mathbf{y}_k, \mathbf{f}_{\theta^\star}(x_k))^\top \mathbf{H}_{\theta^\star}^{-1} \nabla_\theta \tilde{L}_{i,j,c}.$$

We assume $\sum_{(x_k,y_k) \in \mathcal{D}_t} \nabla_\theta \hat{\ell}(\mathbf{y}_k, \mathbf{f}_{\theta^\star}(x_k))^\top \mathbf{H}_{\theta^\star}^{-1} \nabla_\theta \tilde{L}_{i,j,c} \cdot \epsilon_{i,j,c} \leq \alpha$, where $\alpha > 0$. Then, through selecting the samples with $\bar{\phi}_{i,j,c}^{wm}(\mathcal{D}_v) \geq \alpha$, the reweighting influence term $\bar{\phi}_{i,j,c}^{rw}(\mathcal{D}_v) \geq 0$. Then we can get the conclusion that by reweighting the training loss term in $\mathcal{S}_- = \{\bar{\ell}_{i,j,c}(\theta) | i \in [N], j \in [M], c \in [C], \bar{\phi}_{i,j,c}^{rw}(\mathcal{D}_v) > \alpha\}$ from training, we have $\mathcal{L}(\mathcal{D}_t; \theta_\epsilon^\star) - \mathcal{L}(\mathcal{D}_t; \theta^\star) \leq 0$.

## F   Computation detail of inverse Hessian

The estimation of influence score requires the computation of the inverse hessian. The size of hessian matrix is propotional to the number of model parameters, thus directly computing the inverse hessian $\mathbf{H}_{\theta^\star}^{-1}$ is very expensive. For the $\mathbf{H}_{\theta^\star}^{-1}$ involved in the reweighting influence, Equation(11), and the weight-moving influence Equation (14), we employed the LiSSA (Linear time Stochastic Second-Order Algorithm) method [1], which provide an unbiased estimation of the Hessian-vector product through implicitly computing it with a mini-batch of samples. As demonstrated in the previous works [4, 41], the stochastic method is efficient and relatively accurate for sample-wise influence estimation. The algorithm can be summarized as:

- Step 1.  Let $v := \sum_{(x',y') \in \mathcal{D}_v} \nabla_\theta \ell(y', \mathbf{f}_{\theta^\star}(x')) \left( \sum_{(x',y') \in \mathcal{D}_v} \nabla_\theta \ell(y', \mathbf{f}_{\theta^\star}(x')) \right)$ for weight-moving method ), and initialize the inverse HVP estimation $\mathbf{H}_{0,\theta^\star}^{-1} v = v$.

- Step 2.   For $i \in \{1, 2, \ldots, J\}$, recursively compute the inverse HVP estimation using a batch size $B$ of randomly sampled a data point $(x^{i'}, y^{i'})$, $\mathbf{H}_{i,\theta^\star}^{-1} v = v + \left( I - \nabla_\theta^2 \ell(y^{(i')}, \mathbf{f}_{\theta^\star}(x^{(i')})) \right) \mathbf{H}_{i-1,\theta^\star}^{-1} v$, where $J$ is a sufficiently large integer so that the above quantity converges.

- Step 3. Repeat Step 1-2 $T$ times independently, and return the averaged inverse HVP estimations.

For the computation of self-influence, the influence estimation is required for each training sample. For real-world dataset, estimating each sample separately using the LiSSA method is intolerable. Instead of applying the stochastic method for each training sample, we leverage the relation between Hessian matrix and Fisher matrix, and use the K-FAC method [27] directly compute the inverse Hessian matrix. We refer interested readers to Barshan et al. [3] for details regarding the K-FAC approximation for the computation of inverse Hessian.

## G   Experimental details and additional results

### G.1   Dataset statistics and implementation details

We summarize the dataset statistics in Table 6. All of the involved datasets are either publicly available or will be released upon the acceptance of this paper.

All experiments ran on a machine with an Intel(R) Xeon(R) CPU E5-2678 v3 with a 126G memory and a GeForce GTX 1080Ti-11GB GPU.

All the code was implemented in Python and largely based on the WRENCH [50] codebase.

### G.2   Estimate actual effect of LF via source-aware IF

As described in Section 3.3, we can use the source-aware IF calculated by the reweighting method to estimate the influence score of each LF ($\phi_{\lambda^{(j)}} = \sum_{i=1}^{N} \sum_{c=1}^{C} \bar{\phi}_{i,j,c}$, the 2nd row in Table 1). Here,

Table 6: Dataset statistics.

| Domain (↓) | Dataset (↓) | #Label | #LF | #Train Data | #Valid Data | #Test Data |
|---|---|---|---|---|---|---|
| Tabular | Census | 2 | 83 | 10,083 | 5,561 | 16,281 |
| | Mushroom | 2 | 20 | 6481 | 812 | 813 |
| | PW | 2 | 15 | 8654 | 1105 | 1106 |
| | spambase | 2 | 15 | 3595 | 460 | 461 |
| Text | IMDb | 2 | 5 | 20,000 | 2,500 | 2,500 |
| | Yelp | 2 | 8 | 30,400 | 3,800 | 3,800 |
| | Youtube | 2 | 10 | 1,586 | 120 | 250 |
| Image | DN-real | 5 | 5 | 2,587 | 323 | 324 |
| | DN-sketch | 5 | 5 | 1,777 | 222 | 223 |
| | DN-quickdraw | 5 | 5 | 2,000 | 250 | 250 |
| | DN-painting | 5 | 5 | 2,462 | 308 | 308 |
| | DN-infograph | 5 | 5 | 1,213 | 152 | 152 |
| | DN-clipart | 5 | 5 | 773 | 97 | 97 |

we study how well these influence scores reflect the actual effect of each LF. In Table 7, we again report the Spearman's ranking correlation coefficient ($\leq 1$). Please note that the estimated influence of each LF is more likely to be prone to noise since it involves more terms in summation than that of a training data. Also note that here we compare the estimated influence against the actual effect, which is calculated by removing the loss terms associated with each LF and retraining the end model. From the results, we can see that although there do exists cases where the results do not pass the significance test, the averaged ranking correlations are good (from 0.675-0.766). Such observations indicate that source-aware IF could be useful when estimating the influence of each IF.

Table 7: Spearman's ranking correlation coefficient ($\leq 1$) between actual effect of each LF and that estimated from source-aware IF. We highlight results that do not pass the significance test in underline.

| Dataset | Census | Mushroom | PW | Spambase | IMDb | Yelp | Youtube | DN-real | DN-sketch | DN-quickdraw | DN-painting | DN-infograph | DN-clipart | Avg. |
|---|---|---|---|---|---|---|---|---|---|---|---|---|---|---|
| MV | 0.968 | 0.883 | 0.929 | 0.939 | -0.700 | 0.262 | 0.842 | 1.000 | 0.300 | 0.900 | 1.000 | 1.000 | 1.000 | 0.717 |
| DS | 0.801 | 0.534 | 0.950 | 0.939 | 1.000 | 0.810 | 0.830 | 0.800 | 0.400 | 0.400 | 1.000 | 0.700 | 0.800 | 0.766 |
| Snorkel | 0.884 | 0.974 | 0.907 | 1.000 | 1.000 | 0.833 | 0.370 | 0.900 | 1.000 | 0.600 | 1.000 | -0.600 | -0.100 | 0.675 |

## G.3    Additional experiments on neural network

Although when the end model is neural network, the theory of IF breaks down [22], we are curious about whether the proposed source-aware IF is still effective. Thus, we conduct the experiments of identifying mislabeling of LFs (see Table 8) and improving test loss (see Table 9) using two-layer neural network with the ReLU activation function as end model in this section. For simplicity, we do not include the methods based on RelatIF. From the results, we can see that, similar to the experimental results in the main body of this paper, source-aware IF outperforms baselines with a large margin. This indicates that source-aware IF is still effective even when the theory does not hold.

## G.4    The study of label model approximation

In this study, we take a close look at the label model approximation. Specifically, we present the Mean Squared Error (MSE) and expected disagreement (DE) between the output of label model **DS/Snorkel** and their approximated version in Table 10. The MSE is evaluated against the predicted label posterior of models, and the ED is the expectation of $Y_1 \neq Y_2$ over the training samples, where $Y_1$ and $Y_2$ is the predicted label of the label model and its approximated version. From the results, we can see that both metrics have quite low value across datasets for binary classification, while they are relatively larger for multi-class classification, which is because label model for binary classification is much easier to approximate. Even for multi-class classification, all the EDs are still less then 22%, which indicates the approximated label model could replicate most of the predicted labels of original label model.

Table 8: Performance comparison results on identifying mislabeling of LFs. We report the average precision (AP) score averaged over LFs for each dataset. The larger the AP is, the better the method identify mislabeling of LFs.

| Dataset | KNN | MV | | | | DS | | | | Snorkel | | | |
|---|---|---|---|---|---|---|---|---|---|---|---|---|---|
| | | LM | EM | RW | WM | LM | EM | RW | WM | LM | EM | RW | WM |
| Census | 0.810 | 0.809 | 0.787 | **0.854** | 0.824 | 0.787 | 0.787 | **0.789** | 0.788 | 0.787 | 0.787 | **0.805** | 0.803 |
| Mushroom | **0.975** | 0.923 | 0.828 | **0.956** | 0.954 | 0.828 | 0.828 | **0.908** | 0.893 | 0.828 | 0.828 | **0.895** | 0.861 |
| PW | 0.822 | 0.863 | 0.766 | **0.887** | 0.865 | 0.766 | 0.766 | **0.887** | 0.884 | 0.766 | 0.766 | **0.880** | 0.873 |
| Spambase | 0.782 | 0.772 | 0.738 | **0.871** | 0.801 | 0.738 | 0.738 | 0.784 | **0.789** | 0.738 | 0.738 | 0.799 | **0.809** |
| IMDb | 0.702 | 0.767 | 0.699 | **0.786** | 0.773 | 0.699 | 0.699 | **0.771** | 0.761 | 0.699 | 0.699 | 0.732 | **0.732** |
| Yelp | 0.752 | 0.792 | 0.731 | **0.836** | 0.813 | 0.731 | 0.731 | 0.761 | **0.775** | 0.731 | 0.731 | 0.836 | **0.839** |
| Youtube | 0.831 | **0.949** | 0.826 | 0.859 | 0.872 | 0.826 | 0.826 | **0.889** | 0.885 | 0.826 | 0.826 | **0.909** | 0.905 |
| DN-real | 0.711 | 0.447 | 0.417 | **0.906** | 0.878 | 0.417 | 0.417 | **0.573** | 0.536 | 0.445 | 0.417 | **0.746** | 0.651 |
| DN-sketch | 0.321 | 0.339 | 0.316 | **0.730** | 0.665 | 0.316 | 0.316 | **0.490** | 0.465 | 0.316 | 0.316 | **0.424** | 0.421 |
| DN-quickdraw | 0.362 | 0.256 | 0.255 | **0.713** | 0.675 | 0.255 | 0.255 | **0.437** | 0.389 | 0.255 | 0.255 | **0.507** | 0.494 |
| DN-painting | 0.454 | 0.416 | 0.360 | **0.736** | 0.697 | 0.360 | 0.360 | **0.615** | 0.557 | 0.360 | 0.360 | **0.650** | 0.598 |
| DN-infograph | 0.361 | 0.385 | 0.356 | **0.621** | 0.606 | 0.356 | 0.356 | **0.538** | 0.516 | 0.356 | 0.356 | 0.416 | **0.418** |
| DN-clipart | 0.437 | 0.487 | 0.434 | **0.844** | 0.822 | 0.434 | 0.434 | **0.556** | 0.545 | 0.434 | 0.434 | 0.630 | **0.644** |
| Avg. | 0.640 | 0.631 | 0.578 | **0.815** | 0.788 | 0.578 | 0.578 | **0.692** | 0.676 | 0.580 | 0.578 | **0.710** | 0.696 |

Table 9: Performance comparison results on the test loss of end models.

| Dataset | MV | | | | | DS | | | | Snorkel | | | |
|---|---|---|---|---|---|---|---|---|---|---|---|---|---|
| | ERM | IF | GIF | RW | WM | ERM | IF | RW | WM | ERM | IF | RW | WM |
| Census | 0.484 | 0.382 | 0.368 | **0.368** | 0.381 | 0.663 | 0.432 | **0.414** | 0.432 | 0.580 | **0.391** | 0.407 | 0.453 |
| Mushroom | 0.220 | 0.161 | 0.168 | **0.144** | 0.152 | 0.370 | 0.216 | **0.212** | 0.242 | 0.336 | 0.190 | **0.168** | 0.284 |
| PW | 0.394 | **0.309** | 0.372 | 0.334 | 0.338 | 0.477 | 0.318 | **0.316** | 0.322 | 0.487 | **0.333** | 0.358 | 0.333 |
| Spambase | 0.529 | 0.336 | 0.370 | **0.307** | 0.349 | 0.663 | 0.345 | **0.336** | 0.364 | 0.411 | **0.283** | 0.299 | 0.295 |
| IMDb | 0.496 | **0.481** | 0.494 | 0.498 | 0.498 | 0.652 | 0.605 | 0.613 | **0.586** | 0.585 | **0.576** | 0.582 | 0.580 |
| Yelp | 0.524 | 0.352 | 0.463 | 0.394 | **0.344** | 0.460 | **0.373** | 0.447 | 0.452 | 0.513 | **0.406** | 0.497 | 0.490 |
| Youtube | 0.332 | **0.215** | 0.230 | 0.291 | 0.272 | 0.337 | **0.265** | 0.302 | 0.312 | 0.326 | **0.269** | 0.288 | 0.291 |
| DN-real | 1.053 | 0.618 | 0.547 | **0.395** | 0.469 | 0.860 | 0.482 | 0.214 | **0.190** | 0.918 | 0.545 | 0.431 | **0.409** |
| DN-sketch | 1.263 | 0.926 | 1.245 | **0.869** | 0.894 | 1.579 | **1.058** | 1.093 | 1.065 | 1.502 | 1.089 | 0.993 | **0.989** |
| DN-quickdraw | 1.626 | 1.106 | 1.492 | **0.757** | 0.768 | 1.345 | 0.840 | **0.683** | 0.769 | 1.620 | 1.444 | 1.188 | **1.158** |
| DN-painting | 1.242 | 0.876 | 0.947 | **0.755** | 0.797 | 1.043 | 0.680 | **0.615** | 0.654 | 1.210 | 0.872 | 0.915 | **0.857** |
| DN-infograp | 1.477 | 1.251 | 1.169 | 1.238 | **1.180** | 1.207 | **1.198** | 1.282 | 1.269 | 1.504 | 1.490 | 1.418 | **1.336** |
| DN-clipart | 1.084 | 0.913 | 0.830 | **0.741** | 0.779 | 1.000 | **0.901** | 1.042 | 0.993 | 0.944 | 0.676 | 0.729 | **0.624** |
| Avg. | 0.825 | 0.610 | 0.669 | **0.545** | 0.555 | 0.820 | 0.593 | **0.582** | 0.589 | 0.841 | 0.659 | 0.636 | **0.623** |

## G.5 How many mislabelings should be removed to have positive impact on test loss?

Here, we would like to answer the question of how many mislabeling to remove to have positive impact on the test loss. Concretely, this could be reflected by a quantity $\beta$ we call maximal removing portion (MRP). That is, if we remove more than $\beta\%$ top-ranked negatively influential labelings, the resultant test loss after re-training will be larger than original test loss without any removing. In other words, removing any portion of labelings between 0% and $\beta\%$ has a positive impact on the test loss. We take label model MV and our RW method as an example for this study. We found that the MRP $\beta$ is closely correlated to the accuracy of training labels produced by label model (see Figure 3). This is quite intuitive- for high-quality training set, the MRP would be low since we do not have to remove many labelings when training labels are already accurate, while for low-accuracy cases, the MRP would be high since most of the training labels are incorrect so we could remove a large portion of labelings but still be able to improve the test loss.

| Dataset | Metric | Census | Mushroom | PW | Spambase | IMDb | Yelp | Youtube | DN-real | DN-sketch | DN-quickdraw | DN-painting | DN-infograph | DN-clipart | Avg. |
|---|---|---|---|---|---|---|---|---|---|---|---|---|---|---|---|
| DS | MSE | 0.00041 | 0.00000 | 0.00015 | 0.00003 | 0.00033 | 0.00004 | 0.00094 | 0.02127 | 0.02910 | 0.01689 | 0.01620 | 0.01735 | 0.02620 | 0.00992 |
| | ED | 0.00130 | 0.00000 | 0.00150 | 0.00000 | 0.00000 | 0.00000 | 0.00398 | 0.14766 | 0.21159 | 0.18400 | 0.14866 | 0.12861 | 0.15912 | 0.07588 |
| Snorkel | MSE | 0.00097 | 0.00003 | 0.00111 | 0.00011 | 0.00005 | 0.00050 | 0.00024 | 0.00995 | 0.00729 | 0.00370 | 0.01250 | 0.00749 | 0.00992 | 0.00414 |
| | ED | 0.00400 | 0.00031 | 0.00300 | 0.00056 | 0.00011 | 0.00322 | 0.00066 | 0.13993 | 0.10692 | 0.09100 | 0.19659 | 0.04369 | 0.12807 | 0.05524 |

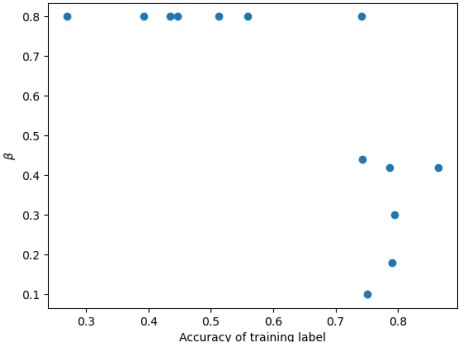

Figure 3: The correlation between maximal removing portion $\beta$ and the accuracy of synthesized training labels.

## G.6 Additional experiments on improving test accuracy / F1

In the main body of the paper, we leverage the proposed method to improve test loss; in this study, we instead present the improved model performance in terms of standard classification metrics, *i.e.*, F1 score and accuracy. Specifically, for binary classification, we adopt F1 score, while for multi-class classification, we use accuracy. For simplicity, we do not include the methods based on RelatIF. From the results, we can draw similar conclusion as Table 4 in the main body, *i.e.*, our methods outperform baselines in most of cases and achieve better averaged performance.

Table 11: Performance comparison results on the classification metrics of end models.

| Dataset | MV | | | | | DS | | | | Snorkel | | | |
|---|---|---|---|---|---|---|---|---|---|---|---|---|---|
| | ERM | IF | GIF | RW | WM | ERM | IF | RW | WM | ERM | IF | RW | WM |
| Census | 0.579 | 0.642 | **0.653** | 0.649 | 0.648 | 0.516 | **0.605** | 0.596 | 0.590 | 0.554 | 0.610 | 0.624 | **0.630** |
| Mushroom | 0.893 | 0.913 | 0.952 | 0.952 | **0.958** | 0.853 | 0.896 | **0.899** | 0.850 | 0.863 | 0.929 | **0.936** | 0.928 |
| PW | 0.844 | 0.876 | 0.873 | 0.877 | **0.878** | 0.799 | 0.866 | 0.863 | **0.870** | 0.807 | 0.867 | 0.867 | **0.875** |
| Spambase | 0.783 | 0.881 | 0.867 | 0.872 | **0.883** | 0.690 | 0.867 | **0.870** | 0.865 | 0.842 | 0.870 | **0.901** | 0.901 |
| IMDb | 0.789 | 0.789 | 0.740 | 0.790 | **0.793** | **0.626** | 0.612 | 0.626 | 0.626 | **0.786** | 0.786 | 0.786 | 0.786 |
| Yelp | 0.839 | 0.839 | **0.847** | 0.842 | 0.833 | 0.853 | 0.853 | **0.862** | 0.850 | 0.850 | 0.850 | **0.852** | 0.845 |
| Youtube | 0.790 | 0.810 | **0.887** | 0.861 | 0.870 | 0.824 | **0.888** | 0.858 | 0.821 | 0.858 | 0.898 | **0.899** | 0.883 |
| DN-real | 0.892 | 0.944 | 0.917 | **0.966** | 0.957 | 0.685 | 0.920 | **0.966** | 0.948 | 0.849 | 0.954 | 0.960 | **0.966** |
| DN-sketch | 0.552 | 0.632 | 0.538 | **0.682** | 0.664 | 0.484 | 0.507 | **0.673** | 0.578 | 0.538 | 0.659 | **0.664** | 0.628 |
| DN-quickdraw | 0.420 | 0.764 | 0.400 | **0.780** | 0.724 | 0.544 | 0.700 | **0.740** | 0.736 | 0.360 | **0.720** | 0.668 | 0.560 |
| DN-painting | 0.656 | 0.821 | 0.763 | 0.818 | **0.860** | 0.695 | **0.847** | 0.831 | 0.847 | 0.614 | 0.815 | **0.854** | 0.834 |
| DN-infograp | **0.612** | 0.586 | 0.566 | 0.526 | 0.599 | 0.553 | 0.559 | **0.579** | 0.546 | 0.539 | 0.520 | 0.546 | **0.579** |
| DN-clipart | 0.691 | 0.711 | 0.691 | **0.742** | 0.732 | 0.639 | 0.691 | **0.711** | 0.670 | 0.701 | **0.804** | 0.794 | 0.784 |
| Avg. | 0.718 | 0.785 | 0.746 | 0.797 | **0.800** | 0.674 | 0.755 | **0.775** | 0.754 | 0.705 | 0.791 | **0.796** | 0.785 |