# OpenReview forum: "Understanding Programmatic Weak Supervision via Source-aware Influence Function"
_NeurIPS.cc/2022/Conference — NeurIPS 2022 Accept_

### Official Review · Reviewer_gUhW · 2022-07-05

**Rating:** 6
**Confidence:** 3
**Soundness:** 3 good
**Presentation:** 3 good
**Contribution:** 3 good

**Summary:**

## Summary
In this work, the authors propose a general framework for quantifying the influence of individual PWS components on the end model. The proposed framework can be used to look into the influence of individual PWS components' effects on the end model. A source-aware influence function is proposed, which leverages the knowledge of the probabilistic label generation process and uses that knowledge to decompose the training loss into multiple terms and eventually individual influence scores. These influence scores are used to quantify the effect of PWS components (source vote, supervision source, and training data).


**Questions:**

n/a

**Strengths And Weaknesses:**

## Strengths
1. The effective use of the source-aware influence function along with the knowledge of the probabilistic label generation process provides a fine-grained analysis tool for the behavior of the end-model.
2. The influence of individual PWS components can be analyzed with the proposed framework.
3. The proposed framework has the capability to work with diverse data domains (tabular, image, textual). Secondly, an ample number of datasets are used in the study.
4. The most responsible LFs as well as their capacity to mislabel can be identified, which is different from identifying the influential training data.
5. The proposed model steers the end-model towards generalization, which is always desirable.
6. The paper is well written.


## Weaknesses
1. The proposed framework hasn’t been shown to work with one-stage methods.
2. Experimental results are reported for two-layer neural networks. It is still to be seen how the proposed method performs with intricate and deep neural architectures.

---

> ### Author Response · Authors · 2022-08-02
> **Author Response to Reviewer gUhW**
>
> Thank you for your valuable feedback!
> Although our method only works for the two-stage method, we think it could inspire future work for studying the effect of components in the one-stage method, where the way of leveraging source votes is more complicated and therefore more challenging.
> As for performance on deep neural architectures, we believe that our work set the foundation for understanding programmatic weak supervision with a more complicated model. We would like to leave those explorations to the future.

---

### Official Review · Reviewer_Dqzp · 2022-07-11

**Rating:** 6
**Confidence:** 2
**Soundness:** 3 good
**Presentation:** 3 good
**Contribution:** 3 good

**Summary:**

This paper introduces a method called source-aware Influence Function (IF) to study the “influence” of individual data, source and class tuples on the performance of different label functions in the programmatic weak supervision paradigm. This differs from previous methods that calculate ordinary IF which can only identify influential training data and not the labeling function or the training data source responsible for mislabeling. The paper introduces a framework for estimating the influence of each training data on the test loss and prediction. The authors introduce two methods to calculate this source-aware IF. They support their method with theorems and proofs. They also show a variety of use-cases on different datasets as to how source-aware IF can improve the performance and understanding of PWS pipelines.

**Questions:**

1. The two methods of calculating source-aware IF correspond to two different ways of perturbing the training loss. How do these 2 differ (maybe through examples) in identifying the most responsible (i,j,c) tuples? From tables 3 & 4, on average, RW seems to be doing better than WM. It would be nice to have a discussion on why that may be.

2. What is the significance of RelatIF here? How is it better/different from ordinary IF and source-aware IF? R-RW seems to be outperforming RW (table 3) for Majority Vote but not for others.

3. It would also be interesting to look at some examples from other datasets (tabular/textual) and see which LFs are responsible. In many real world non-image applications, the LFs may not be created similar to the experimental setup in the paper for the DN datasets.

4. Having identified which LFs have mislabelings, it would be nice to see what can be done about it. For eg, how much does correcting these mislabelings affect the end performance? Is there a minimum number of corrections to be made to improve the classification results? I.e, how many mislabelings need to be identified to positively make an impact on the results?

5. It would also be interesting to look at the difference in classification accuracy/F1 score on an unseen test set with the usual PWS setup and after removing the tuples most responsible for mislabeling (based on a validation set).

6. In table 2, it might be more helpful to have the ‘Misclassified Test Data’ as the first column since the explanation goes from misclassification to the responsible components.

7. There is a small grammatical error in the table 3 caption - “The larger the AP is, the better the method identify mislabeling of LFs.”


**Limitations:**

The authors have mentioned some limitations of their work. The proposed framework may be difficult to implement in the case of more complicated label models. It has not been defined for one-stage PWS pipelines which are becoming more popular recently. The paper does not include experiments on using complicated deep learning architectures as the end model. Additionally, this framework is also limited in its ability to decide/recommend how many samples (responsible for mislabeling) need to be discarded or down weighted for optimal classification performance.

**Strengths And Weaknesses:**

Strengths:
1. The paper tackles a very interesting and important aspect of understanding programmatic weak supervision. The problem is well motivated.

2. The paper includes theoretical proofs of the claims and methods used for developing source-aware IF.

3. The generalization of the generation process of probabilistic labels using eq 6 is nice.

4. The paper includes experiments on a wide range of benchmark datasets and classification tasks.

5. The contributions of the paper are clear, original and significant.

Weaknesses:
1. The motivation for certain aspects of the method and experiments are unclear. For e.g:
         a. The intuition behind why there are two methods for calculating source-aware IF is not clear - why reweighting vs weight-moving. There isn’t a clear study of how one is better or worse than the other and in what cases and why.
         b. The significance of RelatIF in the context of the contributions of this paper is not clear.

2. In the experiments, the training data most responsible for mislabeling is removed and the resulting test loss is reported. However, it is unconvincing if this is the best strategy. What happens if this process is repeated multiple times after removing 1 data point each time or the top n most responsible data points are removed?

3. The test loss is shown to reduce after removing the most negatively influencing tuples, however, there is no mention of how this may translate to classification accuracy or other metrics.

---

> ### Author Response · Authors · 2022-08-02
> **Author Response to Reviewer Dqzp**
>
> Thank you for catching the grammar issue and other comments (Q6 \& Q7)- we have fixed these in our latest version. We answer your other questions as below.
>
> _**Q1 \& W1.a**: The two methods of calculating source-aware IF correspond to two different ways of perturbing the training loss. How do these 2 differ (maybe through examples) in identifying the most responsible (i,j,c) tuples? From tables 3 \& 4, on average, RW seems to be doing better than WM. It would be nice to have a discussion on why that may be._
>
> **R1**: Thank you for asking this question so that we could further explain the difference between RW and WM.
> First, RW has convenient computation only when the $\sigma$ is identity function, while WM is agnostic to the $\sigma$ function, so WM is more general than RW in terms of computation. However, RW has better theoretical performance guarantee than WM, because the WM case of Theorem 1 requires more assumptions to hold than its counterpart of RW. We think that this explains why RW leads to better overall test loss than WM. We will also add some discussion to the paper.
>
> _**Q2 \& W1.b**: What is the significance of RelatIF here? How is it better/different from ordinary IF and source-aware IF?._
>
> **R2**: We include the RelatIF to show that our framework is compatible with state-of-the-art variant of IF method, ie, RelatIF can be combined with the proposed RW and WM, which are based on original IF, leading to R-RW and R-WM. As for the difference between ordinary IF and RelatIF, the latter additionally considers and constraints the global effect of a sample on the model to regularize the effect of some dominating samples. These dominating samples could be outlier and identified as the most influential sample for most of test data, making them poor choices for explaination.
>
> _**Q3**: some examples from other datasets._
>
> **R3**: We added an additional example of Yelp to the appendix (Appendix G.6), a text sentiment classification dataset. And this example leads to similar observations as what we get from the visual example in the main body of the paper.
>
> _**Q4**: How much does correcting the identified mislabelings affect the end performance? Is there a minimum number of corrections to be made to improve the classification results? I.e, how many mislabelings need to be identified to positively make an impact on the results?_
>
> **R4**:
> First, our method, as well as the ordinary IF method, can only identify most negatively influential training samples/labelings, which are then removed from the training to improve the test loss. One could incorporated our method in a human-in-the-loop system to leverage human expertise to correct those samples if they're corrupted.
> As for how many mislabelings need to be removed, we conducted an additional study trying to answer this (see Figure 3 in Appendix G.5).
> We measure the maximal removing portion (MRP) $\beta$. That is, if we remove more than $\beta$\% top-ranked negatively influential labelings, the resultant test loss after re-training will be larger than original test loss without any removing. In other words, removing any portion of labelings between 0\% and $\beta$\% has a positive impact on the test loss. We take label model MV and our RW method as an example for this study. We found that the MRP $\beta$ is closely correlated to the accuracy of training labels produced by the label model. This is quite intuitive- for a high-quality training set, the MRP would be low since we do not have to remove many labelings when training labels are already accurate, while for low-accuracy cases, the MRP would be high since most of the training labels are incorrect so we could remove a large portion of labelings but still be able to improve the test loss.
>
> _**W2**: What happens if this process is repeated multiple times after removing 1 data point each time or the top n most responsible data points are removed?_
>
> **R5**: Yes, the IF scores produced by our method and baselines can all be used in such a iterative manner, while in experiments, we follow existing convention to evaluate the usefulness of IF scores by one-round removing, since the study of best strategy of leveraging IF scores is orthogonal to our study, and we leave it as future work.
>
> _**Q5 \& W3**: the performance of classification accuracy/F1 score on an unseen test set._
>
> **R6**: We conducted new experiments on improving the classification accuracy and F1 score, and included those results in Appendix G.7. From the results, we can draw a similar conclusion as Table 4 in the main body, ie, our methods outperform baselines in most of cases and achieve better averaged performance, which shows that the improvement of tess loss could be translated to that of classification metrics.

---

> > ### Author Response · Authors · 2022-08-02
> > **Results of R6**
> >
> >
> > |              |      |    MV     |        |           |           |           |      |    DS     |           |           |           |      |  Snorkel  |           |           |           |
> > | ------------ | :--: | :-------: | :----: | :-------: | :-------: | :-------: | :--: | :-------: | :-------: | :-------: | :-------: | :--: | :-------: | --------- | :-------: | :-------: |
> > | **Dataset**  |      |  **ERM**  | **IF** |  **GIF**  |  **RW**   |  **WM**   |      |  **ERM**  |  **IF**   |  **RW**   |  **WM**   |      |  **ERM**  | **IF**    |  **RW**   |  **WM**   |
> > | Census       |      |   0.579   | 0.642  | **0.653** |   0.649   |   0.648   |      |   0.516   | **0.605** |   0.596   |   0.590   |      |   0.554   | 0.610     |   0.624   | **0.630** |
> > | Mushroom     |      |   0.893   | 0.913  |   0.952   |   0.952   | **0.958** |      |   0.853   |   0.896   | **0.899** |   0.850   |      |   0.863   | 0.929     | **0.936** |   0.928   |
> > | PW           |      |   0.844   | 0.876  |   0.873   |   0.877   | **0.878** |      |   0.799   |   0.866   |   0.863   | **0.870** |      |   0.807   | 0.867     |   0.867   | **0.875** |
> > | Spambase     |      |   0.783   | 0.881  |   0.867   |   0.872   | **0.883** |      |   0.690   |   0.867   | **0.870** |   0.865   |      |   0.842   | 0.870     | **0.901** |   0.901   |
> > | IMDb         |      |   0.789   | 0.789  |   0.740   |   0.790   | **0.793** |      | **0.626** |   0.612   |   0.626   |   0.626   |      | **0.786** | 0.786     |   0.786   |   0.786   |
> > | Yelp         |      |   0.839   | 0.839  | **0.847** |   0.842   |   0.833   |      |   0.853   |   0.853   | **0.862** |   0.850   |      |   0.850   | 0.850     | **0.852** |   0.845   |
> > | Youtube      |      |   0.790   | 0.810  | **0.887** |   0.861   |   0.870   |      |   0.824   | **0.888** |   0.858   |   0.821   |      |   0.858   | 0.898     | **0.899** |   0.883   |
> > | DN-real      |      |   0.892   | 0.944  |   0.917   | **0.966** |   0.957   |      |   0.685   |   0.920   | **0.966** |   0.948   |      |   0.849   | 0.954     |   0.960   | **0.966** |
> > | DN-sketch    |      |   0.552   | 0.632  |   0.538   | **0.682** |   0.664   |      |   0.484   |   0.507   | **0.673** |   0.578   |      |   0.538   | 0.659     | **0.664** |   0.628   |
> > | DN-quickdraw |      |   0.420   | 0.764  |   0.400   | **0.780** |   0.724   |      |   0.544   |   0.700   | **0.740** |   0.736   |      |   0.360   | **0.720** |   0.668   |   0.560   |
> > | DN-painting  |      |   0.656   | 0.821  |   0.763   |   0.818   | **0.860** |      |   0.695   | **0.847** |   0.831   |   0.847   |      |   0.614   | 0.815     | **0.854** |   0.834   |
> > | DN-infograph  |      | **0.612** | 0.586  |   0.566   |   0.526   |   0.599   |      |   0.553   |   0.559   | **0.579** |   0.546   |      |   0.539   | 0.520     |   0.546   | **0.579** |
> > | DN-clipart   |      |   0.691   | 0.711  |   0.691   | **0.742** |   0.732   |      |   0.639   |   0.691   | **0.711** |   0.670   |      |   0.701   | **0.804** |   0.794   |   0.784   |
> > | Avg.         |      |   0.718   | 0.785  |   0.746   |   0.797   | **0.800** |      |   0.674   |   0.755   | **0.775** |   0.754   |      |   0.705   | 0.791     | **0.796** |   0.785   |

---

### Official Review · Reviewer_Pj3z · 2022-07-17

**Rating:** 6
**Confidence:** 4
**Soundness:** 3 good
**Presentation:** 3 good
**Contribution:** 3 good

**Summary:**

This paper focuses on weakly-supervised classification tasks where (i) multiple weak sources (i.e., labeling rules) are used in a label model (e.g., weighted majority voting) to generate soft labels for unlabeled data; (ii) the soft labels are used to train an end model (e.g., logistic regression, neural classifier) with a smooth cross-entropy loss.

The paper proposes a method to evaluate the influence of each weak source on the end model's performance by considering the choice of the label model, a critical component in this weak supervision pipeline. The main idea behind the proposed method is to decompose the training loss of the end model into multiple components corresponding to the individual weak sources. Then, two techniques are proposed to compute a source-aware influence function, namely reweighting and weight-moving. In cases of label models involving an exponential function in their generation process, the paper applies the proposed source-aware influence function techniques by training an approximate label model based on the identify function.

The paper evaluates and compares the proposed method with various alternatives for multiple use cases and datasets across tabular, text, and image modalities.

**Questions:**

* What is the size of the validation set used in this paper? Does it make sense to use at least a few labeled data per class while training either the label model or the end model?


**Limitations:**

Yes.

**Strengths And Weaknesses:**

Strengths:
* Overall, the paper is clearly written and provides clear and substantiated arguments.
* The paper addresses an interesting and challenging problem. Evaluating the influence of weak sources on the end model's performance is an increasingly important research direction given the increasing adoption of programmatic labeling in both research and industrial settings.
* The proposed method shows promising experimental results on an extensive experimental evaluation of multiple techniques on several scenarios and datasets. It is demonstrated that including the label model into the computation of IF helps across multiple use cases.


Weaknesses:
* It is hard to understand why (in theory) the proposed method should be effective across label model choices.
  *  The proposed method unifies three types of label models into a single equation (Eq. (6) in Section 3.1) without having discussed the inner workings of each label model (e.g., weighted aggregation in Snorkel). Thus, it is hard for a reader not familiar with these label models to understand Eq. (6) and possibly the rest of the method.
  * For cases where $\sigma(\cdot)$ is not the identity function, it is not clear whether the proposed method explains the real influence of each component. The approximated label model (that uses the identity function) is simpler and is not guaranteed to have the same behavior as the original label model. (Indeed, the two label models lead to different rankings as shown in Figure 1. Thus, it is not clear whether the estimated influence scores explain the real influence of each component. In addition to comparing the rankings, it would also help to compare the predicted labels of the original and approximated label models to give an idea of how well the latter approximates the former.
  * It is not clear why the proposed method (resorting into approximations of label models) would in theory be more effective than a simpler method that first computes (source-agnostic) influence scores for each instance and then aggregates instance-level scores into source-level scores.
* The problem addressed in this paper is a simplified (and possibly unrealistic?) case compared to problems addressed in practice in weak supervision
  * According to the problem definition no labeled data are assumed for training, however, labeled data are considered in a validation set. In practice, (at least few) labeled data are considered during training (either in the label model or in the end model) and have been shown to improve the end model's performance. For, example, clean labeled data could be combined with weakly labeled data with weights. It is not clear whether the proposed method can be applied in this setting.

---

> ### Author Response · Authors · 2022-08-02
> **Author Response to Reviewer Pj3z**
>
> We thank you for your detailed and helpful comments! We answered your questions as below.
>
>
>
>
> _**W1.1**: Discussion of the inner workings of each label model._
>
> **R1**: Thanks for point this out! We added a brief explanation of label models in Section 3.1, and in the Appendix A, we also discuss the inner workings of each involved label model, as well as a figure for illustration purpose.
>
> _**W1.2**: Comparison of the predicted labels of the original and approximated label models._
>
>
> **R2**: We evaluated the mean squared error (MSE) and the expected disagreement (ED) of the label model and its approximated version (the results can be found in Table 10 of appendix and we also put it below), where the MSE is operated over the predicted label posterior and the ED is $E[Y_1\neq Y_2]$ over data samples ($Y_1$ and $Y_2$ are predicted labels of label model and its approximation). From the results, we can see that both metrics have quite low value across datasets for binary classification, while they are relatively larger for multi-class classification, which is because label model for binary classification is much easier to approximate. Even for multi-class classification, all the EDs are still less then 22\%, which indicates the approximated label model could replicate most of the predicted labels of original label model.
>
> | Dataset | Metric | Census  | Mushroom |   PW    | Spambase |  IMDb   |  Yelp   | Youtube | DN-real | DN-sketch | DN-quickdraw | DN-painting | DN-infograph | DN-clipart |  Avg.   |
> | :-----: | :----: | :-----: | :------: | :-----: | :------: | :-----: | :-----: | ------: | :-----: | :-------: | :----------: | :---------: | :----------: | :--------: | :-----: |
> |   DS    |  MSE   | 0.00041 | 0.00000  | 0.00015 | 0.00003  | 0.00033 | 0.00004 | 0.00094 | 0.02127 |  0.02910  |   0.01689    |   0.01620   |   0.01735    |  0.02620   | 0.00992 |
> |         |   ED   | 0.00130 | 0.00000  | 0.00150 | 0.00000  | 0.00000 | 0.00000 | 0.00398 | 0.14766 |  0.21159  |   0.18400    |   0.14866   |   0.12861    |  0.15912   | 0.07588 |
> | Snorkel |  MSE   | 0.00097 | 0.00003  | 0.00111 | 0.00011  | 0.00005 | 0.00050 | 0.00024 | 0.00995 |  0.00729  |   0.00370    |   0.01250   |   0.00749    |  0.00992   | 0.00414 |
> |         |   ED   | 0.00400 | 0.00031  | 0.00300 | 0.00056  | 0.00011 | 0.00322 | 0.00066 | 0.13993 |  0.10692  |   0.09100    |   0.19659   |   0.04369    |  0.12807   | 0.05524 |
>
>
>
> _**W1.3**: The advantages of proposed method on approximated label models over a baseline that aggregates instance-level IF score as source-level IF score._
>
> **R3**:
> We would like to point out that when the $\sigma$ is not identity function, the baseline mentioned also relies on the approximated label model to be valid, because in this case each source contributes to an instance via a complex normalization function, eg, softmax. Thus, here the IF score is not addable, and this motivates use to use approximated label model so that the IF score is addable.
> When both methods work on approximated label models, they both suffer from the same approximation error introduced by label model approximation, but in practice, one could use our primitive source-aware IF to remove some of the (possibly harmful) votes of an individual source while the baseline can only be used to remove a source as well as all of its votes.
> Actually, this baseline is the GIF method we included in the Table 4.
>
> _**W2**: Leveraging clean labeled data._
>
> **R4**:
> Yes, one can use clean labeled data and still leverage our method to explain the effect of upstreaming components.
> When we use clean labeled data via an additional loss term, let's say $\ell_{clean}$, we could still compute the source-aware IF in the same way since the new loss $\ell_{clean}$ does not depend on the sources and therefore would not affect our derivation.
> In a sum, using clean labeled data for training better model and our goal of understanding the upstreaming components is orthogonal to each other.
>
>
> _**Q1**: What is the size of the validation set used in this paper?_
>
> **R5**: For all the datasets, we use the standard train/validation/test split and the validation size can be found in Table 6 of the appendix.
>
> _**Q2**: Does it make sense to use at least a few labeled data per class while training either the label model or the end model?_
>
> **R6**: Yes, one can definitely use labeled data to improve the performance, while our method can still be used in these cases to help understand the effect of components on model prediction of new data samples as mentioned above (R4).

---

### Author Response · Authors · 2022-08-02
**Response to all reviewers, thank you!**

We thank all the reviewers for their helpful feedback on the submission!
Based on the valuable comments and suggestions, we have added some new content/experiments (**Appendix G.4-G.7**) to the latest draft (text highlighted in blue). Please note that the full detail for some of the new results are in the appendix rather than the response text below due to length, sorry for the inconvenience!

We’ve provided replies to individual reviewer comments. Please let us know if you have additional questions or need further clarifications. Thanks again!

---

### Meta-Review · Area_Chair_UfPY · 2022-08-30

**Recommendation:** Accept
**Confidence:** Certain

**Metareview:**

This paper proposes source-aware Influence Function (IF) to study the “influence” of individual data, source, and class tuples on the performance of different label functions in the programmatic weak supervision paradigm. The proposed method has the capability to work with diverse data domains (tabular, image, textual). An ample number of datasets are used in the experiments.

The reviewers agree that the proposed method is interesting and sound, the experiments are thorough, and the results provide valuable insights for future work. Reviewers' raised concerns and questions are properly addressed by the author's response.

**Award:**

No

---

### Decision · Program_Chairs · 2022-09-14

Accept